# Charting organellar importomes by quantitative mass spectrometry

Christian D. Peikert[1,*], Jan Mani[2,*], Marcel Morgenstern[1], Sandro Käser[2], Bettina Knapp[1], Christoph Wenger[2], Anke Harsman[2], Silke Oeljeklaus[1], André Schneider[2] & Bettina Warscheid[1,3]

Protein import into organelles is essential for all eukaryotes and facilitated by multi-protein translocation machineries. Analysing whether a protein is transported into an organelle is largely restricted to single constituents. This renders knowledge about imported proteins incomplete, limiting our understanding of organellar biogenesis and function. Here we introduce a method that enables charting an organelle's importome. The approach relies on inducible RNAi-mediated knockdown of an essential subunit of a translocase to impair import and quantitative mass spectrometry. To highlight its potential, we established the mitochondrial importome of *Trypanosoma brucei*, comprising 1,120 proteins including 331 new candidates. Furthermore, the method allows for the identification of proteins with dual or multiple locations and the substrates of distinct protein import pathways. We demonstrate the specificity and versatility of this ImportOmics method by targeting import factors in mitochondria and glycosomes, which demonstrates its potential for globally studying protein import and inventories of organelles.

[1] Faculty of Biology, Department of Biochemistry and Functional Proteomics, Institute of Biology II, University of Freiburg, Schänzlestrasse 1, 79104 Freiburg, Germany. [2] Department of Chemistry and Biochemistry, University of Bern, Freiestrasse 3, CH-3012 Bern, Switzerland. [3] BIOSS Centre for Biological Signalling Studies, University of Freiburg, Schänzlestrasse 18, 79104 Freiburg, Germany. * These authors contributed equally to this work. Correspondence and requests for materials should be addressed to A.S. (email: andre.schneider@ibc.unibe.ch) or to B.W. (email: bettina.warscheid@biologie.uni-freiburg.de).

Most eukaryotic proteins are encoded in the nuclear genome and synthesized in the cytosol, but many need to be further sorted before they reach their final destination. These proteins follow distinct import pathways into various organelles. The resulting protein inventories define the identity and functions of organelles in eukaryotic cells. Knowing the subcellular localization of a protein, its import route, and the complete protein compendium of an organelle therefore lies at the heart of cell biology.

A protein's subcellular localization can be analysed microscopically using fluorescently labelled proteins or antibodies. Alternatively, it can be determined biochemically by cell fractionation. However, while the first method allows the analysis of proteins in living or at least morphologically intact cells, it cannot easily be adopted to proteome-wide studies. Traditional cell fractionation in combination with mass spectrometry (MS), referred to as organellar proteomics[1,2], does not suffer from this drawback. However, it remains essentially impossible to isolate organelles to homogeneity. This considerably limits the accuracy of this approach, which is especially true for the discrimination of low abundant organellar constituents and co-purifying contaminants. Both approaches have also severe limitations when dealing with proteins that are simultaneously localized to more than one compartment. To alleviate some of these restrictions, elaborate proteomic profiling approaches employing quantitative MS have been implemented[3–11].

In previous work, we used a global quantitative profiling method to establish a most complete list of mitochondrial outer membrane (OM) proteins of *Trypanosoma brucei*[7]. In the present study, we sought to establish a map of the entire mitochondrial proteome of *T. brucei* that is imported from the cytosol. With this aim, we designed and implemented a method, which combines the power of cell fractionation, stable isotope labelling (that is, SILAC[12], peptide stable isotope dimethyl labelling[13]) and quantitative MS with an additional filter that specifically selects for proteins whose mitochondrial localization depends on the main protein translocation machinery located in the mitochondrial OM. This filter is based on inducible knockdown of the central, pore-forming subunit of the translocase to efficiently impair mitochondrial protein import *in vivo*. We demonstrate that this method overcomes issues of limited specificity inherent to existing organellar proteomics approaches as it provides information about whether a protein is imported or not. It even allows for the unambiguous identification of mitochondrial proteins with dual or multiple localizations. Moreover, the method holds the potential of globally identifying the substrates of different protein import pathways into mitochondria as demonstrated by targeting ATOM40, SAM50 and ERV1. In a proof of concept study, we further show its applicability to the mapping of proteins imported into glycosomes. We believe that this versatile method will enable scientists to address many of the still open questions related to organellar proteomes.

## Results

**Loss of ATOM40 diminishes levels of substrate proteins**. We recently elucidated function and molecular architecture of the archaic translocase of the mitochondrial outer membrane (ATOM), which is essential for the import of mitochondrial precursor proteins into the mitochondrion of the parasitic protozoan *T. brucei*[14,15]. Here we exploit the fundamental role of ATOM as main gatekeeper of mitochondria to develop a method that permits to globally identify the pool of mitochondria-imported proteins to which we refer to as the 'mitochondrial importome'. To effectively impair import of

mitochondrial precursor proteins, we targeted ATOM40, which is the central, pore-forming β-barrel protein of ATOM[16] (Fig. 1a). Following induction of RNA interference (RNAi)-mediated knockdown of ATOM40, we observed two major effects when analysing total cell lysates. First, unprocessed precursor forms of some presequence-containing proteins accumulated (Fig. 1b, CoxIV, mtHSP70, REAP1). As expected when protein import is inhibited, these were localized in the cytosol (Fig. 1c). Second, the steady-state levels of all mitochondrial proteins tested decreased considerably (Fig. 1b), irrespective of their submitochondrial localization and whether they contain a presequence or not. In contrast, the levels of the cytosolic protein EF1a, which serves as control for a non-imported protein, did not change.

We demonstrate that the observed reduction in the amounts of mitochondrial proteins is not due to a change in their respective messenger RNA levels (Fig. 1d). The simplest explanation for these results is that ablation of ATOM40 causes a decrease in the levels of imported proteins along with an accumulation of non-imported, nuclear-encoded precursors in the cytosol. However, the cytosolic accumulation is not always evident since mistargeted proteins are rapidly degraded by the proteasome, as evidenced by their slower degradation in the presence of the proteasome inhibitor MG-132 (Fig. 1e). A similar effect has recently been described for yeast[17]. Thus, the decrease of the mitochondrial levels of imported proteins is mainly due to reduced import. Yet, differences in the extent of reduction are observed as not all imported proteins have the same half-lives in mitochondria. The levels of imported proteins that are subunits of protein complexes, for example, may even be further reduced by intramitochondrial degradation if the levels of putative binding partners become limiting. Nevertheless, a reduction in the mitochondrial abundance of a protein in induced ATOM40-RNAi cells is indicative of its mitochondrial localization.

**Defining the mitochondrial importome**. We envisioned that the quantitative proteomic comparison of gradient-purified mitochondria from induced ( + Tet) versus uninduced ( − Tet) ATOM40-RNAi cells is an effective approach to delineate the mitochondrial importome (Fig. 2a). With the exception of OM proteins that do not rely on ATOM40 for membrane insertion and mitochondrially encoded proteins, this importome represents the entire mitochondrial proteome. The method, which we termed 'ImportOmics', combines the advantages of cell fractionation (that is, reduction of sample complexity and enrichment of low abundant organellar proteins) with a functional filter for specifically selecting proteins imported into mitochondria. Most importantly, this filter is essentially independent of the quality of the cell fractionation.

To test our approach, we first established a small-scale purification protocol that allows for the isolation of mitochondria by differential and Nycodenz density-gradient centrifugation[18] from ∼$10^{10}$ cells of the procyclic form of *T. brucei* (Supplementary Fig. 1a). Second, we performed SILAC of ATOM40-RNAi cells followed by RNAi induction (Fig. 2a and Supplementary Fig. 1b). To minimize indirect effects of ATOM40 knockdown in living cells, induced ATOM40-RNAi cells were harvested early, at the onset of the growth arrest (3 days after induction) (Supplementary Fig. 1b). Uninduced ATOM40-RNAi cells were cultured in parallel and harvested at the same point in time. We observed a comparable growth defect, a significant reduction of the RNAi target ATOM40, and an accumulation of COXIV precursor in all four replicates (Supplementary Fig. 1b,c). Gel-based liquid chromatography mass spectrometry (LC–MS) analyses of mitochondria purified from SILAC-ATOM40

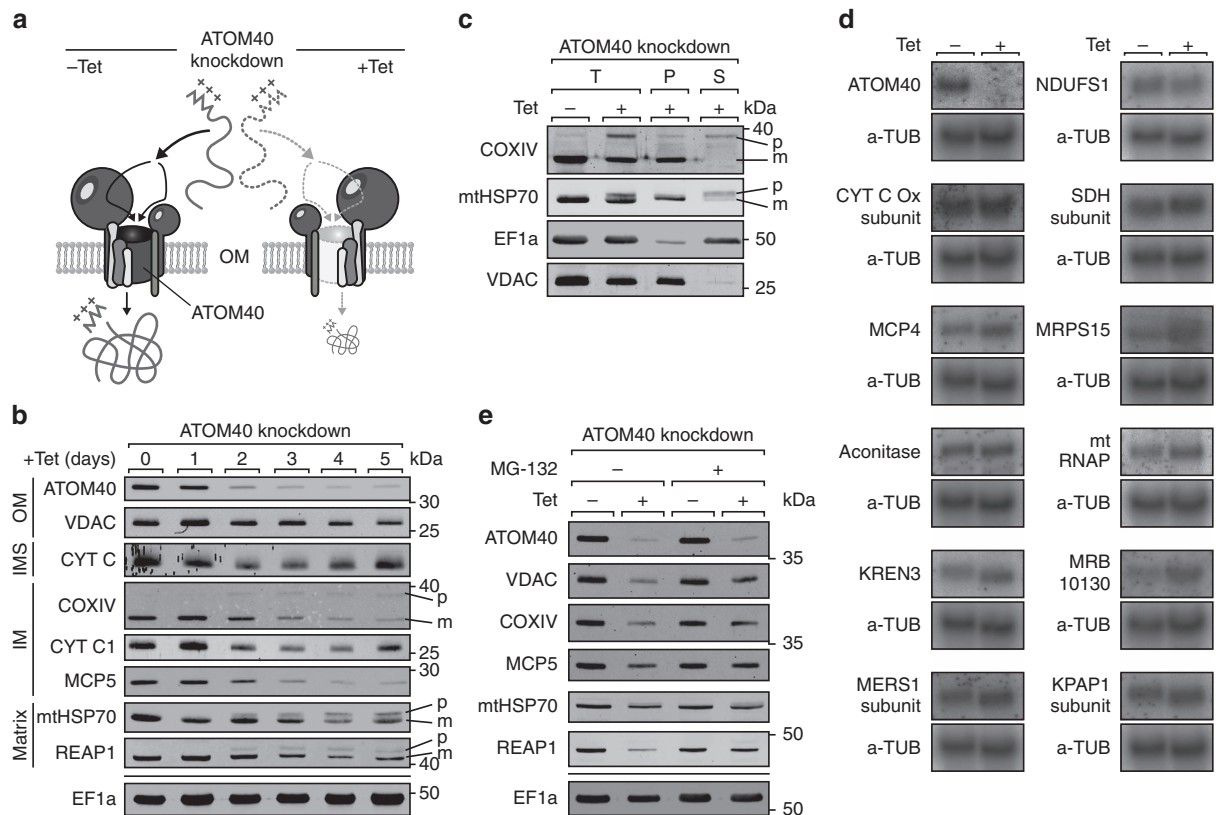

**Figure 1 | Knockdown of ATOM40 reduces mitochondrial protein abundance.** (**a**) Trypanosomal precursor proteins enter the mitochondrion through the ATOM translocase located in the outer membrane (OM). Tetracycline (Tet)-induced RNAi-mediated knockdown (+ Tet) of the central protein import pore ATOM40 impairs import of precursor proteins into the organelle resulting in the depletion of mitochondrial proteins. (**b**) Immunoblots showing the steady-state levels of mitochondrial proteins in whole-cell extracts on Tet-induced RNAi of ATOM40. Proteins are: voltage-dependent anion channel (VDAC), cytochrome c (CYT C), cytochrome c oxidase subunit 4 (COXIV), cytochrome c1 (CYT C1), mitochondrial carrier protein 5 (MCP5), mitochondrial heat shock protein 70 (mtHSP70), and RNA-editing-associated protein 1 (REAP1). Precursors (p) and mature (m) protein forms are indicated. Cytosolic elongation factor 1 alpha (EF1α) serves as a control. Submitochondrial locations of proteins are indicated. IM, inner membrane; IMS, intermembrane space; OM, outer membrane. (**c**) Immunoblot analysis of whole cell (T) and digitonin-extracted, mitochondria-enriched pellet (P) and soluble (S) fractions of uninduced (− Tet) and induced (+ Tet) ATOM40-RNAi cells. Precursor (p) and mature (m) protein forms are indicated. EF1a and VDAC serve as cytosolic and mitochondrial marker protein, respectively. (**d**) Northern blot analysis of mRNAs, isolated from uninduced (− Tet) and induced (+ Tet) ATOM40-RNAi cells, coding for known mitochondrial proteins, which were strongly downregulated on ablation of ATOM40 (see Supplementary Data 1). Alpha-tubulin (α-TUB) serves as loading control. (**e**) Immunoblot analysis of mitochondrial proteins in whole-cell extracts from uninduced (− Tet) and induced (+ Tet) ATOM40-RNAi cells cultured in the presence or absence of the proteasome inhibitor MG-132. Cytosolic EF1a serves as control.

knockdown and control cells (Fig. 2a) resulted in the identification of 2,564 proteins exhibiting at least one SILAC ratio with an overlap of 2,133 proteins (83.2%) across all replicates (Supplementary Fig. 2a, Supplementary Data 1). SILAC ratios were highly reproducible between experiments with Pearson correlation coefficients in the range between 0.85 and 0.93 (Supplementary Fig. 2b). For further evaluation, we considered 2,376 proteins (92%) to be reliably quantified (for details, see 'Methods'). Of these, 934 (39.3%) were part of the mitochondrial reference proteome defined in this work (Supplementary Fig. 2c, Supplementary Data 2a). As depicted in the volcano plot (Fig. 2b), most mitochondrial reference proteins were decreased in abundance, demonstrating that their import was distinctly impaired following ATOM40 ablation. These data highlight the important role of ATOM40 for maintaining the mitochondrial proteome by enabling import of nuclear-encoded mitochondrial proteins.

We showed that loss of ATOM40 leads to the destabilization of the whole ATOM complex[15]. In line with this, SILAC analysis of mitochondria deficient for ATOM40 (10.3-fold reduction)

revealed strongest effects on the two core subunits ATOM14 (9.7-fold red.) and ATOM11 (9.0-fold red.), the former protein stabilizing the core complex and the latter promoting the association of the core with its receptor subunits ATOM46 and ATOM69 (ref. 15). Hence, downregulation of the receptors ATOM46 (6.0-fold) and ATOM69 (4.6-fold) is presumably a consequence of the loss of ATOM11. Interestingly, ATOM12 (2.4-fold red.), exhibiting an antagonistic function to ATOM11, is least affected in addition to the recently identified new subunit ATOM19 (ref. 14) (2.6-fold red.).

Another, more specialized translocase in the mitochondrial OM of trypanosomes is the sorting and assembly machinery (SAM) facilitating folding and insertion of β-barrel proteins into the OM[19,20]. The pore-forming subunit of SAM is the highly conserved SAM50, which itself is a β-barrel protein and requires ATOM40 for import into mitochondria[19]. Accordingly, SAM50 levels were considerably reduced (3.4-fold) in mitochondria lacking ATOM40 (Fig. 2b).

To exclude the possibility that inhibition of mitochondrial protein import may indirectly cause the depletion of non-mitochondrial

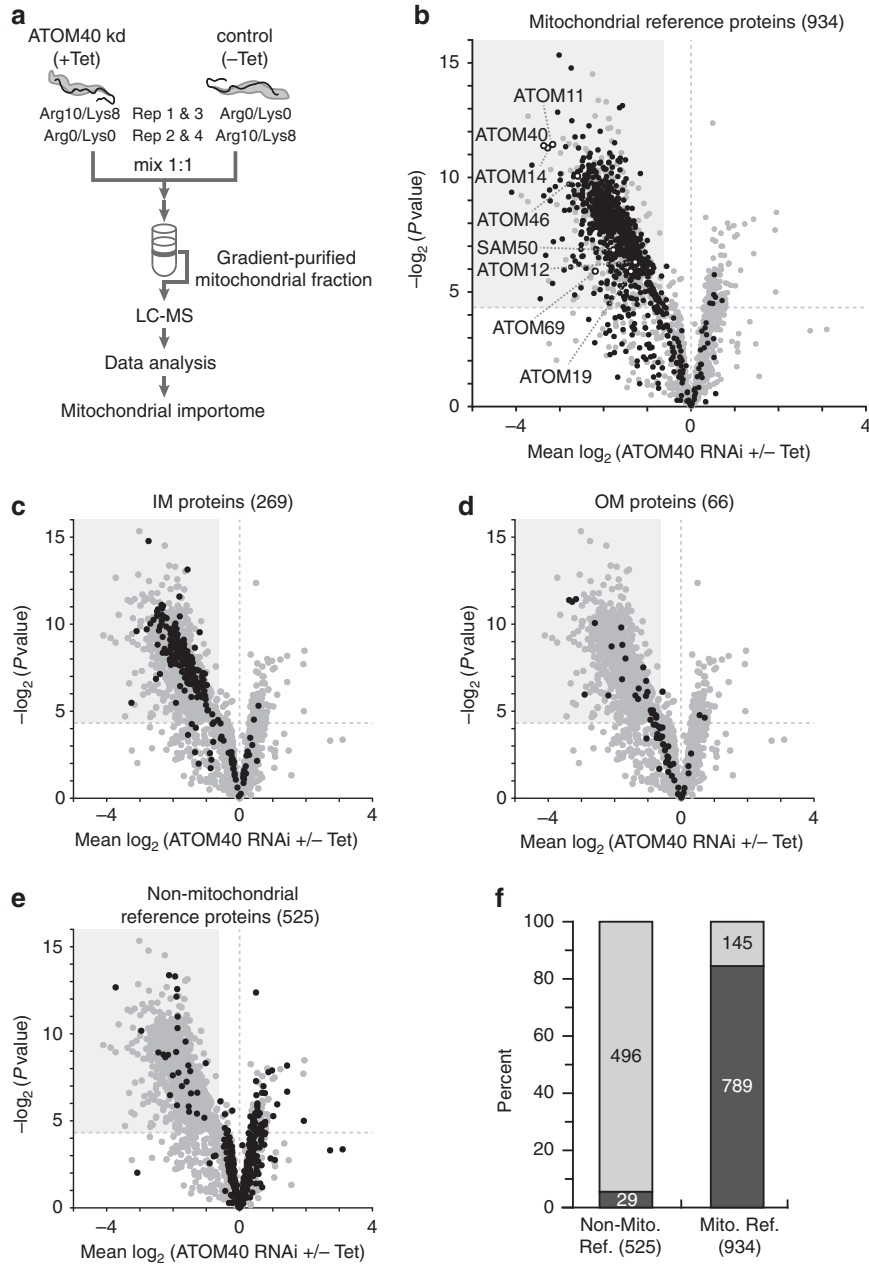

**Figure 2 | Delineation of the ATOM40-dependent mitochondrial importome. (a)** General workflow to study the mitochondrial importome. Using stable isotope labelling by amino acid in cell culture (SILAC), procyclic ATOM40-RNAi cells were differentially labelled with heavy arginine (Arg10) and lysine (Lys8) or their light counterparts (Arg0/Lys0). Tetracycline-induced ( + Tet) and uninduced ( − Tet) ATOM40-RNAi cells were mixed at day 3 after induction and mitochondria were purified by differential and Nycodenz density gradient centrifugation ($n = 4$). Mitochondria were analysed using a gel-based LC-MS approach followed by computational data analysis to establish the mitochondrial importome. **(b)** SILAC-RNAi data obtained in **a** were visualized in a volcano plot. Proteins with a $P$ value < 0.05 ($n = 4$) and a mean $\log_2$ ratio of less than or equal to − 0.62 were considered significantly decreased in abundance (shaded area) and represent the here established mitochondrial importome. Subunits of the ATOM complex and the β-barrel protein SAM50 are annotated. Mitochondrial reference proteins (934) are depicted in black, others (1,442) in grey. **(c–e)** Volcano plot shown in **b** with different classes of proteins highlighted in black as indicated. For details about mitochondrial inner membrane (IM) and outer membrane (OM) proteins as well as the definition of non-mitochondrial reference proteins, see 'Methods' and Supplementary Data 1 and 2. **(f)** Distribution of 525 non-mitochondrial reference proteins and 934 mitochondrial reference proteins quantified in gradient-purified mitochondrial fractions. For each reference set, the portion of proteins that are part of the mitochondrial importome (mean $\log_2$ ratio less than or equal to − 0.62, $P$ value < 0.05, $n = 3$) is depicted in dark grey, while the fraction of proteins that do not belong to the importome is depicted in light grey.

proteins that would potentially be detected as false positive hits in an ImportOmics experiment if co-purified with mitochondria, we quantitatively analysed whole lysates of induced ( + Tet) versus uninduced ( − Tet) ATOM40-RNAi cells to a depth of 5,144 proteins (Supplementary Data 3). In the SILAC–MS analysis of these whole-cell extracts, we determined 523 proteins with a minimum fold-change of 1.5 ($P$ value < 0.05, $n = 3$). Of these, 515 proteins were reduced in abundance of which the RNAi target ATOM40 and its direct partner proteins ATOM14 and ATOM11 were most affected, while only eight proteins were

upregulated. We further analysed the distribution of 918 mitochondrial and 1,160 non-mitochondrial reference proteins detected in this experiment and found the steady-state levels of 411 mitochondrial but only three non-mitochondrial reference proteins to be reduced in abundance by at least 1.5-fold (P value <0.05, n = 3; Supplementary Fig. 3a). Thus, based on our proteome-wide SILAC analysis of whole-cell extracts, we conclude that the depletion of ATOM40 specifically diminished the mitochondrial proteome but not the cellular proteome.

To demonstrate the specificity of our ImportOmics method, we compared the effects of RNAi-mediated knockdown of ATOM40 and SAM50 by SILAC–MS analyses of organelle-enriched fractions prepared from RNAi cells by single-step extraction using digitonin[21]. In contrast to the extensive decrease of the mitochondrial proteome following ATOM40 ablation (Supplementary Fig. 3b, Supplementary Data 4), knockdown of SAM50 at an early time point only resulted in a specific depletion of its β-barrel substrates VDAC, the VDAC-like protein Tb927.11.10780 and ATOM40 together with its closely associated partner ATOM14 (Supplementary Fig. 3c, Supplementary Data 5). The data also show the scalability of the method as it was combined with a fast and simple fractionation step requiring only a low number (<10[8]) of cells.

To eventually define a high-confidence mitochondrial importome, we determined significance thresholds for SILAC ratios of our data set from gradient-purified mitochondria (Fig. 2b) using the two sets of mitochondrial and non-mitochondrial reference proteomes (Supplementary Fig. 4a,b, Supplementary Data 2a,b). The validity of this approach is shown by the clear separation of the two reference sets (Supplementary Fig. 4c). As a result, we established here the mitochondrial importome comprising 1,120 proteins (Fig. 2b, shaded area). The remaining 1,256 proteins largely represent co-purifying contaminants in gradient-purified mitochondrial fractions.

We achieved high coverage of the mitochondrial inner membrane (IM) (269/282 proteins, 95.4%) and OM constituents (66/81, 81.5%)[7] in our mitochondrial fractions (Fig. 2c,d, Supplementary Data 1). Of note, the previously determined IM and OM protein compositions were obtained from highly purified mitochondrial subfractions[7], whereas the source for this study were isolated mitochondria. As expected, not all OM constituents (25/66, 37.9%) are part of the importome (Fig. 2d). Biogenesis of some trypanosomal integral OM proteins does not require ATOM40, but depends on pATOM36 (ref. 22). Also in yeast, several OM proteins with one or more transmembrane α-helical segments are inserted independently of Tom40 (ref. 23). Some require the mitochondrial import complex[24], while others may insert spontaneously or even use as yet unidentified insertases[23].

Most importantly, though, the discriminative power of our ImportOmics method is clearly demonstrated by the distribution of non-mitochondrial reference proteins with 496 out of 525 (94.5%) not being affected following ATOM40 knockdown (Fig. 2e,f, Supplementary Data 1) Thus, only 29 (5.5%) of the non-mitochondrial reference proteins, but 789 (84.5%) of the mitochondrial ones are part of the newly established importome (Fig. 2f), underscoring the effectiveness of our method for the global characterization of the imported mitochondrial proteome.

**Identification of 331 new mitochondrial candidate proteins**. We explored our importome data set for new mitochondrial constituents and identified 331 proteins for which a mitochondrial localization has not been reported (Fig. 3a, Supplementary Data 6). Justified by the virtual absence of obvious contaminants (for example, cytoplasmic ribosomal, flagellar and cytoskeletal proteins) in our mitochondrial importome, we deemed these proteins 'new mitochondrial candidate proteins'. To validate our

data, we selected five candidates and further six proteins with a reported, but so far unconfirmed mitochondrial localization, tagged their C termini using the c-Myc epitope and expressed them in procyclic T. brucei cells. Immunofluorescence analysis showed that all tagged proteins co-localized with ATOM40 (Fig. 3b, Supplementary Fig. 5), thus corroborating our importome data.

In yeast, ~60–70% of all mitochondrial precursor proteins possess an N-terminal targeting signal[23]. We therefore searched for the presence of putative N-terminal presequence segments in the 331 new mitochondrial candidates using the algorithms TargetP and MitoFates. N-terminal presequences were predicted for 152 new candidates (46%) (Fig. 3c, Supplementary Data 6). In comparison, 845 proteins (68.3%) of our mitochondrial reference set contain putative targeting sequences (Supplementary Data 2a), which is in accordance with yeast data[23]. We here propose that many of the newly identified mitochondrial candidate proteins may likely possess internal or so far uncharacterized targeting signals. In fact, only 4 of the 11 newly validated mitochondrial proteins contain a predicted N-terminal presequence (Supplementary Data 6).

To obtain insight into potential functions, we classified new mitochondrial candidates based on Gene Ontology (GO) slim terms in the domains 'biological process', 'cellular component' and 'molecular function' (Fig. 3d, Supplementary Data 6). However, only little information is available as most proteins have so far not been characterized. For candidates with annotations, the most frequent terms found were 'amino-acid metabolism', 'other metabolic processes' and 'oxidoreductase activity'. Furthermore, we performed large-scale protein BLAST analysis to identify functional orthologues in yeast, mouse or human (Supplementary Data 7). We retrieved BLAST hits for 91 new mitochondrial candidates and of these, 46 had orthologues in all three species (Fig. 3e), indicating that they are evolutionary conserved. In fact, all 46 orthologues were recently localized to human mitochondria[9] and 42 to mitochondria of mouse embryonic stem cells[10].

To conclude, ImportOmics enabled the discovery of 331 new mitochondrial candidate proteins, thereby expanding the known mitochondrial proteome of T. brucei by more than 25%. We believe that this rich dataset will trigger many follow-up studies leading to a better understanding of mitochondrial biology.

**Aminoacyl-tRNA synthetases localize to mitochondria**. We reasoned that the ImportOmics method should allow for the identification and localization of mitochondrial proteins that do not exclusively reside in mitochondria but exhibit dual or multiple subcellular locations. In T. brucei, the mitochondrial genome is devoid of tRNA genes. This lack is compensated for by import of a small fraction of essentially all cytosolic tRNAs[25]. Thus, with very few exceptions, the same set of tRNAs is present in both the cytosol and the mitochondrion. To function in translation, each tRNA needs to be charged with the correct amino acid by its corresponding aminoacyl-tRNA synthetase (aaRS). T. brucei has 23 distinct aaRSs, 17 of which are encoded by single-copy genes, while three are encoded by two genes (LysRS-1/-2, AspRS-1/-2 and TrpRS-1/-2), corresponding to the cytosolic or mitochondrial version of the enzyme. Thus, we would assume that all of the 17 aaRS encoded by single-copy genes are dually localized.

We attempted to demonstrate the postulated dual localization for the trypanosomal tyrosyl-tRNA synthetase (TyrRS) using two different biochemical fractionation methods. Figure 4a illustrates that in a digitonin extraction of whole cells, the C-terminally c-Myc-tagged TyrRS cofractionates with the cytosolic marker

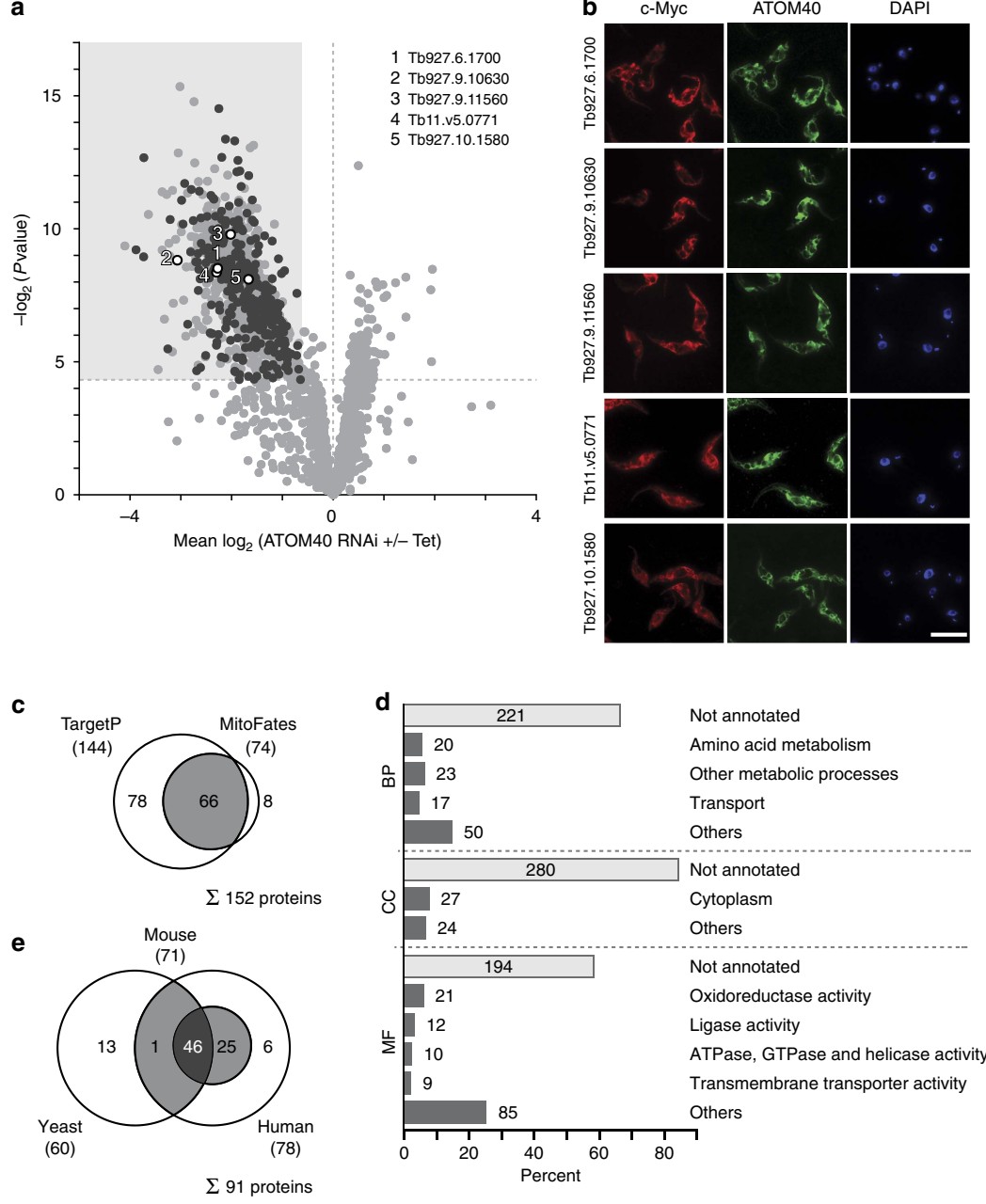

**Figure 3 | New mitochondrial candidate proteins. (a)** Volcano plot as shown in Fig. 2b highlighting the 331 newly identified mitochondrial candidate proteins in black. Validated new mitochondrial candidate proteins (see **b**) are annotated by gene IDs as shown. All other proteins are depicted in light grey. (**b**) Immunofluorescence microscopy analysis of selected mitochondrial candidate proteins in *T. brucei*. Candidates were expressed as C-terminally tagged c-Myc fusion proteins (red) and stained using anti-Myc antibodies. Mitochondria were visualized using anti-ATOM40 serum (green), nuclear and mitochondrial DNA were stained with 4,6-diamidino-2-phenylindole (DAPI) (blue). Scale bar, 10 μm. (**c**) Prediction of mitochondrial targeting sequences for new mitochondrial candidate proteins. Shown are the total numbers and the overlap of candidates with predicted N-terminal signal peptides according to TargetP and MitoFates. (**d**) Functional classification of new mitochondrial candidate proteins according to GO slim terms in the domains 'biological process' (BP), 'cellular component' (CC) and 'molecular function' (MF). The number of proteins assigned to a given term is indicated. (**e**) Overlap of human, mouse and yeast proteins identified as homologues of the new mitochondrial candidate proteins.

EF1a. The same observation is made when gradient-purified mitochondria are analysed (Fig. 4b). From these results, we would conclude that TyrRS is exclusively cytosolically localized. ImportOmics, however, leads to a different conclusion. Using this approach, we could demonstrate the mitochondrial localization of 15 aaRSs. These comprise the mitochondria-specific enzymes LysRS-2, AspRS-2 and TrpRS-2, while the others are predicted to be dually localized. (Fig. 4c, Supplementary Data 1).

The fact that these aaRSs are specifically downregulated in gradient-purified mitochondrial fractions on knockdown of ATOM40 excludes them to be cytosolic contaminants. In line with this, LysRS-1, which is known to be a cytosol-specific enzyme[25], while still detected in small amount in isolated mitochondria, was not affected following ATOM40 depletion (Fig. 4c, Supplementary Data 1). Thus, ImportOmics allows for the reliable assignment of mitochondrial proteins with dual or

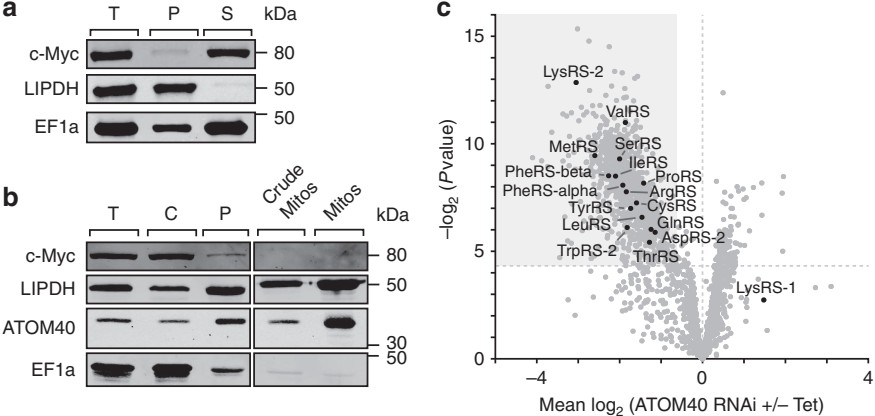

**Figure 4 | Identification of the mitochondrial pool of dually localized aminoacyl-tRNA synthetases. (a)** Immunoblot analysis of whole-cell extract (T) and digitonin-extracted, mitochondria-enriched pellet (P) and soluble (S) fractions of cells expressing C-terminally c-Myc-tagged tyrosyl-tRNA synthetase. Dihydrolipoyl dehydrogenase (LipDH) and EF1a serve as mitochondrial and cytosolic marker, respectively. **(b)** Immunoblot analysis of fractions containing equal amounts of proteins collected during isolation of mitochondria from the cell line analysed in Fig. 1a. T, whole-cell extract; C, cytosol; P, pellet after cell lysis; crude mitos, crude mitochondria before gradient; mitos, gradient-purified mitochondria. LipDH and ATOM40 serve as mitochondrial markers and EF1a as cytosolic marker. **(c)** Distribution of aminoacyl-tRNA synthetases quantified in the SILAC-based analysis of mitochondria isolated from induced ( + Tet) and uninduced ( − Tet) ATOM40-RNAi cells. Same plot as in Fig. 2c with aminoacyl-tRNA synthetases marked in black.

multiple localizations of which only minute amounts are present in mitochondria.

**Identification of putative substrates of the MIA pathway.** Our ImportOmics method provides the great potential for globally identifying substrates of different import factors as indicated by our SILAC data from ATOM40-RNAi and SAM50-RNAi experiments (see Fig. 2b and Supplementary Fig. 3). To further explore this intriguing possibility, we decided to target the mitochondrial disulfide relay system involved in the import of the majority of soluble intermembrane space (IMS) proteins. In yeast and human, this pathway, which also mediates the oxidative folding of its substrates, is termed mitochondrial intermembrane space assembly (MIA) pathway and requires the oxidoreductase MIA40 and the sulfhydryl oxidase ERV1/ALR[26]. Most of the so far characterized MIA pathway substrates in yeast and human are small and rich in cysteines that are arranged in either twin-Cx3C or -Cx9C motives (with x being any amino acid except cysteine). The double cysteine motifs stabilize their typical helix–loop–helix core structure by formation of disulfide bonds. Typical representatives of these classical MIA substrates are the tiny TIM proteins, for example, TIM9 and TIM10 (ref. 27). Additional substrates of this pathway such as Tim22 (refs 26,28) are more complex and thus less easily identified.

In *T. brucei*, the MIA pathway is unusual as this organism expresses an orthologue of ERV1 but lacks a MIA40 (refs 29,30). Whether ERV1 alone is sufficient to mediate import of small IMS proteins or whether another unknown protein replaces MIA40 function is presently unknown. In addition, experimental evidence for trypanosomal MIA pathway substrates is lacking. Thus, to obtain novel insight, we chose ERV1 as RNAi target in procyclic *T. brucei* cells (Fig. 5a). Induction of RNAi-mediated knockdown of ERV1 resulted in a gradual decrease of the steady-state levels of TIM9 (ref. 31; Fig. 5b), the human and yeast orthologues of which are bona fide MIA pathway substrates[32]. In contrast, the steady-state levels of the soluble IMS protein cytochrome c (CYT C) and the cytosolic protein EF1a remained unchanged. In yeast, import of cytochrome C into mitochondria was proposed to occur independently of Erv1/Mia40 (ref. 28), which is in line with our data.

To globally identify MIA pathway substrates, we performed ImportOmics. In our experiments, induced ERV1-RNAi cells were harvested before onset of the growth phenotype (2.5 days after induction; Supplementary Fig. 6a). We quantified proteins in mitochondria-enriched fractions of induced ( + Tet) versus uninduced ( − Tet) ERV1-RNAi cells by MS using peptide stable isotope dimethyl labelling[13]. The analysis resulted in the quantification of 2,820 proteins, 2,520 (90%) of which were quantified in each biological replicate with reproducible ratios between the experiments (Supplementary Fig. 6b,c, Supplementary Data 8). As depicted in Fig. 5c, 25 proteins were significantly reduced in abundance by at least twofold (P value < 0.05, n = 3) and, thus, were classified as putative MIA pathway substrates. Among these were five tiny TIM proteins that we previously detected in our recent study of the trypanosomal IM protein import machinery[33]. We compiled a list of MIA substrate-like proteins (Supplementary Data 9a) and predicted classical MIA pathway substrates in *T. brucei* through a global search for proteins with two or more Cx3C or Cx9C motifs that are characteristic for mitochondrial IMS proteins (Supplementary Data 9b). Such predicted MIA pathway substrates were clearly overrepresented in the candidate list derived from our ERV1-ImportOmics study in comparison to all quantified proteins in the same experiment (Fig. 5d). Thus, we have globally identified the substrates of the MIA pathway. The results obtained in this analysis demonstrate the validity and the effectiveness of ImportOmics for the identification of *in vivo* substrates of distinct protein import pathways of organelles.

**Identification of proteins imported into glycosomes.** So far, we successfully applied our ImportOmics method to mitochondria and showed that it identifies mitochondrial proteins regardless of whether they exclusively localize to mitochondria or not. Moreover, the method holds great potential for globally studying substrates of different import factors as demonstrated by targeting ATOM40, SAM50 and ERV1 of the same organelle. To show that the method can also be applied to organelles other than mitochondria, we selected the glycosome.

The glycosome is a specialized peroxisome-like organelle of trypanosomatids, which harbours several enzymes of glycolysis[34]

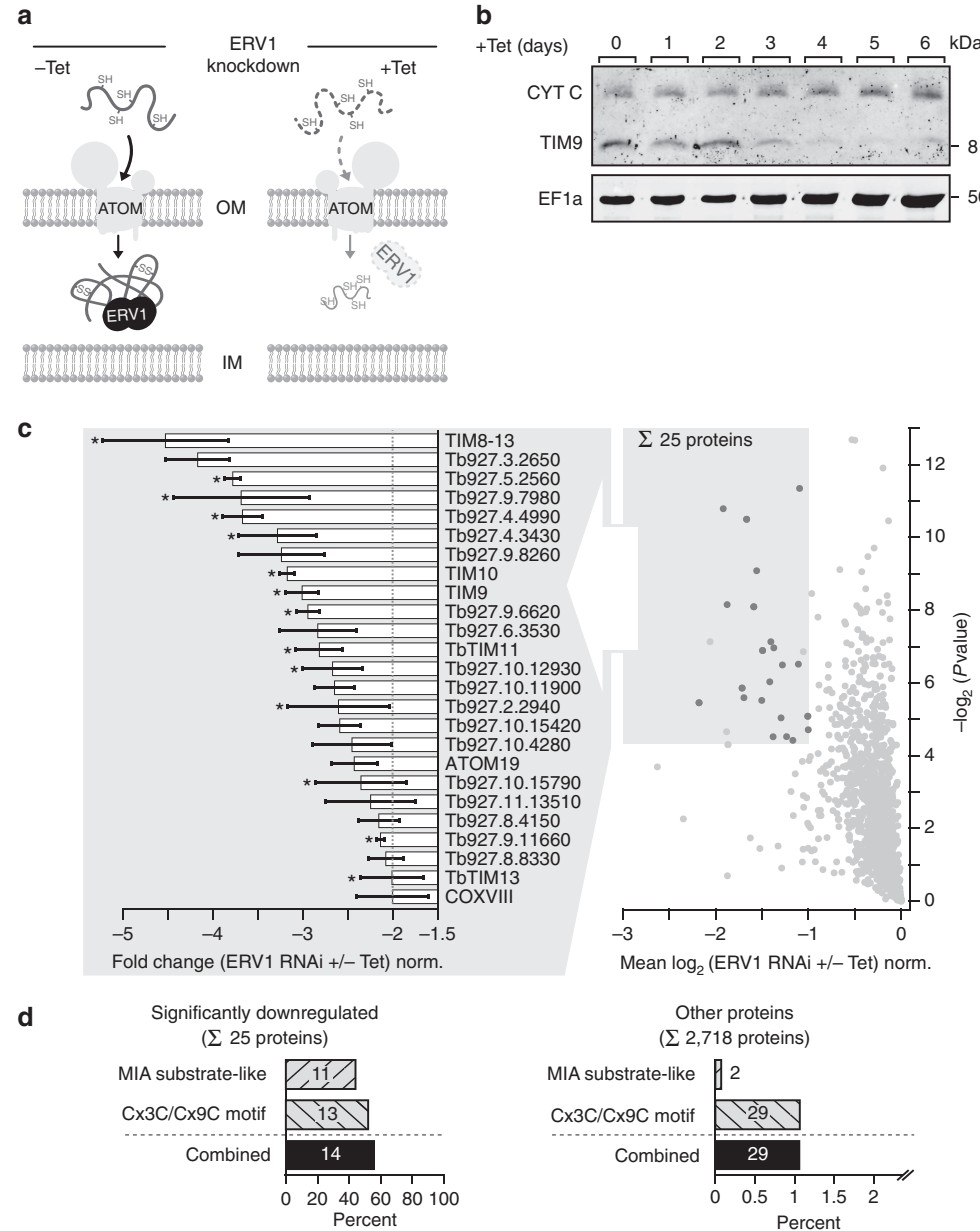

**Figure 5 | Identification of putative MIA pathway substrates by ablation of the sulfhydryl oxidase ERV1.** (**a**) A subset of nuclear-encoded soluble proteins located in the mitochondrial intermembrane space (IMS) is imported via the mitochondrial IMS assembly (MIA) pathway. Import and correct folding of these cysteine-rich proteins requires the sulfhydryl oxidase ERV1 ( − Tet). Tetracycline-induced RNAi-mediated knockdown of ERV1 results in impaired import and depletion of MIA pathway substrates in the IMS ( + Tet) and their subsequent degradation in the cytosol. (**b**) Immunoblot analysis of whole-cell extracts showing the steady-state levels of the IMS proteins cytochrome c (CYT C) and TIM9 as well as the cytosolic protein EF1a following tetracycline-induction ( + Tet) of ERV1 RNAi for the indicated time. (**c**) Proteins reduced in abundance in mitochondria-enriched fractions of induced ERV1-RNAi cells. MS-based quantification of proteins from induced ( + Tet) versus uninduced (–Tet) cells was based on peptide stable isotope dimethyl labelling ($n = 3$). Proteins with a mean $\log_2$ ratio of less than or equal to $-1$ (corresponding to a fold change of less than or equal to $-2$) and a $P$ value $<0.05$ (shaded area) were considered significantly reduced in abundance on ERV1 knockdown. Right graph: dark grey dots, significantly reduced proteins also present in the mitochondrial importome. Left graph: error bars indicate the s.e.m.; *predicted MIA pathway substrates. (**d**) Putative MIA pathway substrates predicted by similarity to known MIA substrates in other organisms (MIA substrate-like) and a general Cx3C/Cx9C motif-based approach are highly overrepresented among the set of proteins significantly downregulated following ERV1 knockdown (left) compared to the set of all other proteins quantified in this experiment (right). Shown is the percentage of putative MIA pathway substrates predicted by the aforementioned approaches in each subset.

and components of other metabolic pathways including the β-oxidation of fatty acids and the pentose–phosphate pathway[34]. As known for peroxisomes, proteins destined for the matrix are targeted to the glycosome either by a C-terminal peroxisomal targeting sequence (PTS) 1 or an N-terminal PTS2 followed by transport across the membrane mediated by multi-protein complexes[34]. Work in yeast has shown that PTS1- and PTS2-containing proteins are imported into peroxisomes using two distinct import pores, which share the peroxin PEX14 (ref. 35). Furthermore, RNAi experiments indicate that the PEX14 orthologue of trypanosomes is essential for growth and survival of procyclic- and bloodstream-form parasites in

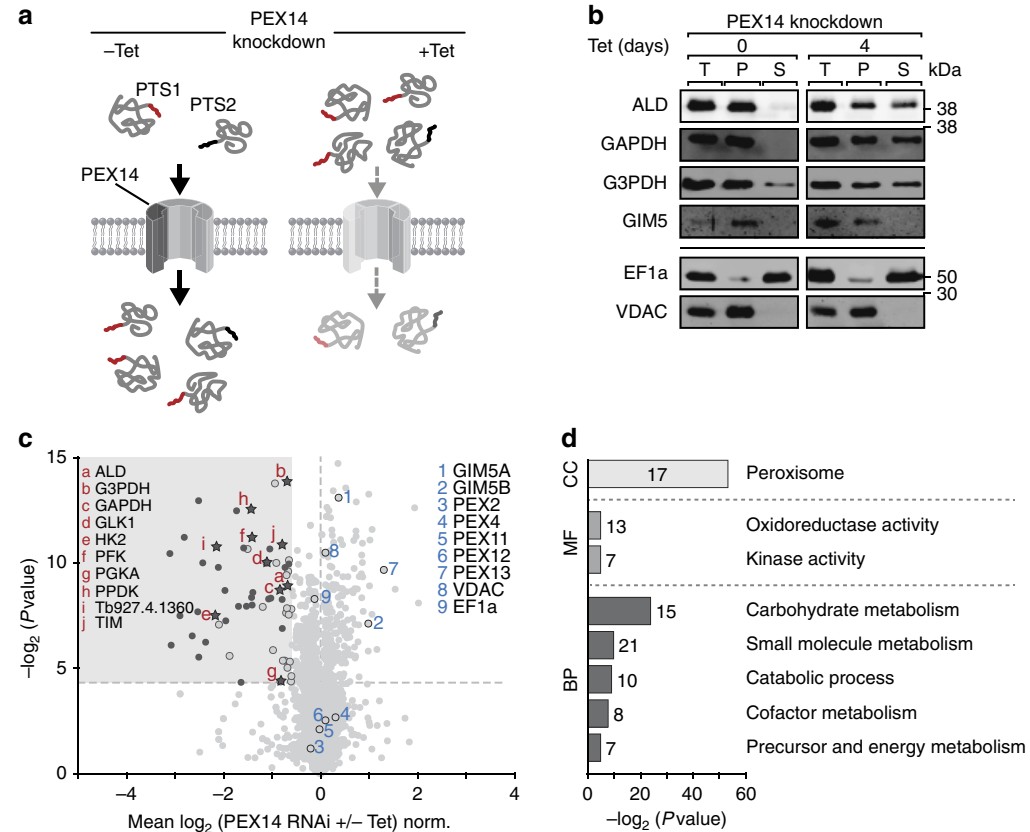

**Figure 6 | Targeting PEX14 to reveal proteins imported into glycosomes. (a)** PTS1- and PTS2-dependent import of nuclear-encoded glycosomal matrix proteins requires PEX14 ( − Tet), a core component of the glycosomal protein import machinery. Inducible RNAi-mediated knockdown of PEX14 ( + Tet) prevents protein import into glycosomes and leads to an accumulation of import substrates in the cytosol over time. PTS, peroxisomal targeting sequence. **(b)** Immunoblot analysis of whole cells (T), glycosome-containing pellet (P) and soluble (S) fractions of uninduced ( − Tet, 0 days) and induced ( + Tet, 4 days) PEX14-RNAi cells. Shown are the abundance levels of the glycosomal matrix proteins fructose bisphosphate aldolase (ALD), glyceraldehyde 3-phosphate dehydrogenase (GAPDH), and glycerol 3-phosphate dehydrogenase (G3PDH) and the glycosomal integral membrane protein 5 (GIM5) in the different samples. EF1a and VDAC serve as cytosolic and mitochondrial marker protein, respectively. **(c)** Volcano plot of SILAC-PEX14-RNAi data. PEX14-RNAi cells were differentially labelled using SILAC including a label switch. Subsequently, the glycosome-containing fractions from induced ( + Tet) and uninduced ( − Tet) PEX14-RNAi cells were analysed by LC-MS ($n = 3$). Proteins with a mean $\log_2$ ratio of less than or equal to $− 0.59$ and $P$ value $< 0.05$ ($n = 3$) were considered significantly decreased in abundance following PEX14 knockdown (shaded area). Stars, known glycosomal proteins involved in glycolysis annotated by protein name or gene ID; dark grey dots, further glycosomal proteins affected by PEX14 knockdown; light grey dots, other proteins. Glycosomal membrane proteins including GIM5A, GIM5B and several peroxins (PEX proteins) as well as the mitochondrial membrane and the cytosolic marker protein VDAC and EF1a, respectively, are highlighted. norm., normalized. **(d)** GO term enrichment analysis of proteins with a significantly decreased abundance in glycosome-containing pellet fractions following PEX14 knockdown. $P$ values after Benjamini–Hochberg false discovery rate (FDR $< 0.05$) correction were plotted against their corresponding GO terms from the three main domains 'cellular component' (CC), 'molecular function' (MF) and 'biological process' (BP). The number of proteins assigned to a given term is indicated.

glucose-containing medium[36]. Hence, we used an inducible PEX14-RNAi cell line of procyclic *T. brucei* to impair glycosomal matrix protein import (Fig. 6a). In this cell line, PEX14 is efficiently depleted causing a growth arrest when cells are grown on glucose (Supplementary Fig. 7a). Total cell extract, cytosolic and crude glycosome-containing pellet fractions obtained by differential centrifugation of uninduced and induced PEX14-RNAi cells were analysed by immunoblotting (Fig. 6b). The results show that following PEX14 depletion the levels of glycosomal matrix proteins (ALD, GAPDH, G3PDH) were reduced in the pellet fraction, but increased in the cytosol. We therefore concluded that non-imported glycosomal proteins remain stable in the cytosol, most likely because they are folded before import. In contrast, GIM5, an abundant, dimeric glycosomal integral membrane protein[37], was not affected in its localization and abundance following PEX14 knockdown

(Fig. 6b). This finding confirms that glycosomal membrane proteins do not depend on PEX14 for import[34].

To show the applicability of our ImportOmics method to glycosomes, we conducted a proof-of-concept study in which we analysed the effects of PEX14 knockdown by SILAC–MS. As for the immunoblot analysis in Fig. 6b, we did not purify glycosomes, but used instead the glycosome-containing pellet fractions from induced and uninduced PEX14-RNAi cells. LC–MS analyses resulted in 2,032 protein identifications (with at least one SILAC ratio) with an overlap of 1,726 proteins (84.9%, $n = 3$; Supplementary Fig. 7b, Supplementary Data 10). SILAC ratios were consistent between experiments (Supplementary Fig. 7c). The volcano plot shown in Fig. 6c depicts the distribution of SILAC ratios of 1,961 proteins identified in these fractions (Supplementary Data 10). Based on the distribution of glycosomal and non-glycosomal reference proteins (Supplementary Data 11),

61 proteins were found to be distinctly decreased in abundance (mean $\log_2$ ratio less than or equal to $-0.59$, P value $< 0.05$, $n = 3$). Our SILAC data corroborate immunoblotting results (Fig. 6b, pellet fraction). Levels of glycosomal enzymes involved in glycolysis were significantly decreased following PEX14 ablation, while glycosomal membrane proteins or non-glycosomal proteins were not affected (Fig. 6c, Supplementary Data 10). The majority of the downregulated proteins (35 proteins, 57.4%) are annotated as glycosomal proteins and 12 of these comprise a putative PTS1 according to PSORT II (Fig. 6c, Supplementary Data 10). Consistently, GO term enrichment analysis revealed a strong overrepresentation of the terms 'peroxisome' and 'carbohydrate metabolism' as expected for glycosomes (Fig. 6d, Supplementary Data 12).

Using a crude organellar fraction in which glycosomes were present in very low amount (that is, estimated abundance of $< 10\%$ based on MS intensities), we were able to cover 60% of the known glycosomal proteome[38]. We expect that the use of fractions specifically enriched for glycosomes allows for a more in-depth analysis of the PEX14-dependent glycosomal importome and may shed light on the question whether alternative import pathways exist. It is also important to mention that the population of glycosomes (or peroxisomes) is rather heterogeneous with only a part of this population being import competent[39]. In addition, glycosomes are highly dynamic organelles which likely import different sets of enzymes depending on their metabolic state and the environmental condition[39]. Our ImportOmics method may provide here an effective approach to reveal potential differences in the import capacity and matrix protein composition of glycosomes at different physiological conditions.

## Discussion

The ImportOmics method opens new avenues for globally studying protein import into organelles through the combination of RNAi, subcellular fractionation and quantitative MS. The method can be adapted to different cellular protein import systems by targeting a distinct, central component of their protein import machinery. It is a versatile method that can be used in any other organism in which inducible knockdown of a distinct component of an organellar protein import machinery is feasible. We see advantages over other approaches as the method provides specific information about which proteins are imported into the organelle including the detection of proteins that are only partially localized in this membrane-surrounded cellular compartment. The method is highly specific and applicable to organelles that cannot be purified to homogeneity, although we see that the depth of the established importome increases with the purity of the organelle. We envision that the method offers many opportunities for studying organelles that represent endpoints of protein trafficking routes including mitochondria-related organelles (for example, hydrogenosomes, mitosomes), peroxisomes, chloroplasts and other plastids or the nucleus. Many of these organelles are difficult to purify and ImportOmics may therefore be especially suited. Most importantly, it holds great potential for globally identifying *in vivo* substrates of various protein import pathways, for example, through ablation of individual import receptor proteins or distinct components of different protein import machines within the same organelle. In addition to mitochondria and glycosomes as highlighted in this work, ImportOmics should be in particular useful for the study of chloroplasts and other plastids. In plastids, ImportOmics may allow for assigning substrates to the different thylakoid protein import systems. In plant and algae systems, we can envision comparative ImportOmics experiments targeting mitochondria and plastids.

We hope that the method will be widely used in cell biology research to further advance our knowledge of how cell organelles master the challenge to maintain and depend on a proteome that is synthesized in the surrounding cytosol.

## Methods

**Transgenic cell lines.** Transgenic cell lines are based on the procyclic *T. brucei* strain 29-13 (ref. 40) and were grown at 27 °C in SDM-79 (ref. 41) supplemented with 10% (v/v) fetal calf serum. For SILAC experiments, cells were grown in modified SDM-80 (ref. 42) containing 5.55 mM glucose, 15% (v/v) dialysed fetal calf serum (BioConcept, Switzerland), and either the unlabelled 'light' or stable isotope-labelled 'heavy' versions of arginine (1.1 mM; Arg0 or Arg10) and lysine (0.4 mM; Lys0 or Lys8) (Cambridge Isotope Laboratories, USA). Induction of RNAi with tetracycline for 3 days (ATOM40 and SAM50) or 4 days (PEX14) was started 2–3 days after beginning SILAC labelling of the cells.

Inducible RNAi-mediated knockdowns of ATOM40 (Tb927.9.9660), PEX14 (Tb927.10.240) and ERV1 (Tb927.9.6060) were performed using pLEW100-derived stemloop vectors, in which the phleomycine resistance gene had been replaced with a blasticidine or, in the case of ERV1, a puromycine resistance gene, and which allow for ligation of inserts in opposing directions separated by a 460 bp spacer fragment using BamHI/XhoI and HindIII/XbaI restriction sites[16,40,43]. The SAM50 (Tb927.3.4380) RNAi cell line has been described before[44]. For inducible triple c-Myc-tagging of tyrosyl-tRNA synthetase (Tb927.7.3620) and candidate proteins (Tb11.v5.0963, Tb927.6.1700, Tb927.11.3940, Tb927.10.8010, Tb927.3.3680, Tb927.9.10630, Tb927.6.2070, Tb11.v5.0771, Tb927.8.580, Tb927.9.11560 and Tb927.10.1580), the full-length open reading frames were cloned into modified pLEW100 (ref. 40) expression vectors, in which the phleomycine resistance gene had been replaced with a puromycine resistance gene and a triple c-Myc cassette had been inserted for C-terminal positioning of the tag using BamHI, HindIII or AgeI restriction sites[15]. A list of oligonucleotide primers is provided in Supplementary Data 13.

**Isolation of mitochondria.** Mitochondria of high purity were isolated from differentially SILAC-labelled tetracycline-treated and untreated ATOM40-RNAi cells mixed in a 1:1 ratio based on cell counts. The isolation was performed under isotonic conditions essentially as described before[21] with the protocol being adapted to lower cell numbers. Briefly, a total of $\sim 1.5 \times 10^{10}$ cells were collected by centrifugation (1,400 r.c.f., 4 °C, 10 min), washed in SBG buffer (22 mM glucose, 150 mM NaCl, 20 mM sodium phosphate buffer pH 7.9), and resuspended in SoTE buffer (20 mM Tris–HCl pH 8.0, 2 mM EDTA pH 8.0, 600 mM sorbitol). Cells were lysed by $N_2$ cavitation (55 bar, 30 min). After centrifugation (23,400 r.c.f. max, 4 °C, 10 min), the resulting pellet was resuspended in SoTE buffer and treated with DNase (30 min on ice). To remove intact cells and large debris, the lysate was centrifuged (500 r.c.f., 4 °C, 10 min) and the pellet was discarded. Following a further centrifugation step (23,400 r.c.f. max, 4 °C, 10 min), the pelleted material was mixed with 50% Nycodenz and loaded at the bottom of a Nycodenz step gradient (28% 3 ml, 25% 2.5 ml, 21% 2.5 ml, 18% 2.5 ml). After ultracentrifugation (125,000 r.c.f. max., 4 °C, 45 min), mitochondrial vesicles were isolated from the gradient and washed in SoTE buffer prior to mass spectrometric analysis. The experiment was performed in four independent replicates including a label-switch.

Gradient-purified mitochondria were also prepared from cells grown under standard conditions and expressing C-terminally triple c-Myc-tagged tyrosyl-tRNA synthetase[21]. For immunoblot analysis, samples were taken at various steps during the isolation procedure.

**Digitonin fractionation.** Cells were collected by centrifugation (1,400 r.c.f., 4 °C, 10 min) and washed with PBS. Plasma membranes were lysed by incubating the cells in SoTE buffer containing 0.015% (w/v) digitonin (5 min on ice). Subsequent differential centrifugation (6,800 r.c.f., 4 °C, 5 min) yielded an organelle-enriched pellet fraction and a fraction enriched for cytosolic proteins[43]. In case of SILAC experiments, differentially labelled tetracycline-treated and untreated RNAi cells were mixed in equal amounts prior to digitonin treatment. Organelle-enriched fractions of SILAC-labelled ATOM40-, SAM50- or PEX14-RNAi cells were each prepared in three independent replicates including a label-switch. Organelle-enriched fractions of unlabelled ERV1-RNAi cells ($+ / -$ Tet) were generated 2.5 days post induction with tetracycline.

**SDS–PAGE and tryptic in-gel digestion.** Gradient-purified mitochondria isolated from SILAC-labelled ATOM40-RNAi cells ($n = 4$) as well as organelle-enriched fractions generated from SILAC-labelled ATOM40- or PEX14-RNAi cells ($n = 3$) were subjected to SDS–polyacrylamide gel electrophoresis (PAGE) prior to LC–MS analysis. To this end, proteins of gradient-purified mitochondria from ATOM40-RNAi cells were acetone-precipitated and resuspended in 1% (w/v) SDS/0.1 N NaOH; organelle-enriched fractions from ATOM40- and PEX14-RNAi cells were resuspended in urea buffer (8 M urea/50 mM $NH_4HCO_3$). Proteins (30 μg per sample) were separated on 4–12% NuPAGE BisTris gradient gels (Life Technologies) according to the manufacturer's instructions. Proteins were visualized using colloidal Coomassie Brilliant Blue and gel lanes cut into 12 slices.

Following reduction of cysteine residues with 5 mM Tris(2-carboxy-ethyl)phosphine dissolved in 10 mM NH$_4$HCO$_3$ (incubation for 30 min at 37 °C) and subsequent alkylation of free thiol groups with 50 mM iodoacetamide per 10 mM NH$_4$HCO$_3$ (30 min at room temperature in the dark), proteins were in-gel digested using trypsin (37 °C, overnight).

**Proteolytic in-solution digestion.** SILAC-labelled intact cells of ATOM40-RNAi experiments (referred to as 'whole cells') mixed in a 1:1 ratio ($+/-$ Tet) as well as organelle-enriched fractions from SAM50- and ERV1-RNAi cells were resuspended in urea buffer. Proteins were reduced and alkylated as described above. The alkylation reaction was quenched by adding dithiothreitol at a final concentration of 20–33 mM. For tryptic digestion, urea concentration was adjusted to 1.6–2 M with 50 mM NH$_4$HCO$_3$ and proteins were digested overnight at 37 °C. Proteins of ATOM40-RNAi whole cells were pre-digested with LysC in 6 M urea (37 °C, 4 h) followed by tryptic digestion as described.

**Stable isotope dimethyl labelling.** Peptides of organelle-enriched fractions from tetracycline-induced and noninduced ERV1-RNAi cells were differentially labelled using 'light' formaldehyde (CH$_2$O; Sigma-Aldrich) and sodium cyanoborohydride (NaBH$_3$CN; Sigma-Aldrich) or the 'heavy', deuterated versions thereof (CD$_2$O/NaBD$_3$CN; Sigma-Aldrich). Prior to labelling, peptides were desalted using StageTips and vacuum-dried. For labelling, peptides corresponding to 5 µg (replicate 1) or 10 µg of protein (replicates 2 and 3) were resuspended in 100 µl of 100 mM tetraethylammonium bicarbonate, 4 µl of 4% (v/v) CH$_2$O/CD$_2$O, and 4 µl of 0.6 M NaBH$_3$CN/NaBD$_3$CN and incubated for 1 h at 20 °C and 800 r.p.m. Labelling reactions were stopped by adding 16 µl of 1% NH$_3$. Samples were subsequently acidified with 8 µl of 100% formic acid (FA). Differentially dimethyl-labelled peptides of tetracycline-treated and untreated cells were mixed, purified using StageTips, and dried in vacuo. The experiment was performed in three independent replicates including a label-switch.

**High pH reversed-phase fractionation.** Peptides of ATOM40-RNAi whole cells obtained from 300 µg of protein per replicate ($n = 3$) were fractionated by high-pH reversed-phase liquid chromatography[45]. To this end, peptides were desalted using C18 cartridges (3M Empore, St Paul, USA) and dried in vacuo. Peptides were resuspended in 99% solvent A (10 mM ammonium hydroxide, pH 10) and 1% solvent B (10 mM ammonium hydroxide in 90% ACN, pH 10) and loaded onto a C18 Gemini-NX column (150 mm × 2 mm inner diameter, particle size 3 µm, pore size 110 Å; Phenomenex, Torrence, USA) using an Ultimate 3000 HPLC system (Thermo Fisher Scientific, Dreieich, Germany) with a flow rate of 200 µl min$^{-1}$ and a column temperature of 40 °C. Peptides were eluted with a gradient of 1–61% solvent B (starting after 5 min) in 60 min followed by 61–78% B in 2 min and 3 min at 78%. One-minute fractions were collected (starting after 1.5 min, ending at 70.15 min) in a concatenated manner such that every 17th fraction was combined resulting in a total of 16 fractions.

Dimethyl-labelled peptides of ERV1-RNAi experiments were fractionated by high-pH reversed-phase using StageTips. Peptides were resuspended in 10 mM NH$_4$OH, loaded onto StageTips equilibrated with 10 mM NH$_4$OH, and eluted step-wise with 10, 13, 15.6, 18.7, 40.5 and 72% ACN diluted in 10 mM NH$_4$OH resulting in seven fractions including the flow-through.

**LC–MS analysis.** Dried peptide samples were resuspended in 0.1% TFA and analysed by nano-HPLC-ESI-MS/MS on a Q Exactive or an Orbitrap Elite instrument (Thermo Fisher Scientific, Bremen, Germany) each connected to an UltiMate 3000 RSLCnano HPLC system (Thermo Fisher Scientific, Dreieich, Germany). The RSLC systems were equipped with PepMap C18 precolumns (5 mm × 300 µm inner diameter; Thermo Scientific) for washing and preconcentration of the peptides and C18 reversed-phase nano LC columns (Acclaim PepMap RSLC columns; 50 cm × 75 µm inner diameter; particle size 2 µm, pore size 100 Å; Thermo Scientific) for peptide separation at 40 °C and a flow rate of 250 nl min$^{-1}$. Peptides from gradient-purified mitochondria of ATOM40-RNAi cells, analysed on a Q Exactive, were separated using a binary solvent system consisting of 4% dimethylsulphoxide (DMSO)/0.1% FA (solvent A) and 30% ACN/48% methanol/4% DMSO/0.1% FA (solvent B). Peptides were eluted with a gradient of 1–65% solvent B in 30 min followed by 65–99% B in 5 min and 5 min at 99% B. The same solvent system was used for Orbitrap Elite MS analyses of whole cells and organelle-enriched fractions obtained from ATOM40-RNAi experiments as well as ERV1-RNAi samples. LC gradients were as follows: 1–65% B in 60 min, 65–95% B in 5 min, 5 min at 95% B (ATOM40 RNAi, whole cells), 5–65% B in 50 min, 65–95% B in 5 min, 5 min at 95% B (ATOM40 RNAi, organelle-enriched fractions) or 1–65% B in 50 min, 65–95% B in 5 min, 5 min at 95% B (ERV1 RNAi). Separation of peptides derived from PEX14-RNAi cells, analysed on a Q Exactive, was performed using 1.5% DMSO/0.1% FA (solvent A) and 86% ACN/1.5% DMSO/0.1% FA (solvent B) with a gradient of 3–39% B in 30 min, 39–95% B in 5 min and 5 min at 95% B. For Orbitrap Elite MS measurements of peptides from SAM50-RNAi samples, analysed in two technical replicates, an LC gradient ranging from 5–57% solvent B (B: 30% ACN/50% methanol/0.1% FA; A: 0.1% FA) in 265 min followed by 57–99% B in 50 min and 5 min at 99% B was applied.

MS instruments, equipped with a Nanospray Flex ion source with DirectJunction and stainless steel emitters (Thermo Scientific), were externally calibrated using standard compounds. Orbitrap Elite parameters were as follows: MS scans, $m/z$ 370–1,700; resolution ($R$), 120,000 (at $m/z$ 400); automatic gain control (AGC), $1 \times 10^6$ ions; and max fill time, 200 ms. A TOP15 (ATOM40 RNAi, organelle-enriched fraction) or TOP25 (ATOM40 RNAi, whole cells; SAM50- and ERV1-RNAi samples) method was applied for low-energy collision-induced dissociation of multiply charged peptides in the linear ion trap applying a normalized collision energy of 35%, an activation q of 0.25, an activation time of 10 ms, an AGC of $5 \times 10^5$, and a max fill time of 150 ms. The dynamic exclusion time for previously selected precursor ions was set to 45 s. For the Q Exactive, parameters were as follows: MS scan range, $m/z$ 375–1,700; $R$, 70,000 (at $m/z$ 200); AGC, $3 \times 10^6$ ions; max fill time, 60 ms; TOP15-higher-energy collisional dissociation of multiply charged peptides; AGC, $1 \times 10^5$; max fill time (orbitrap), 120 ms; normalized collision energy, 28%; dynamic exclusion time, 45 s.

**MS data analysis.** For protein identification and quantification, MS raw data were processed using MaxQuant[46] (version 1.5.2.8) and its integrated search engine Andromeda[47]. Data derived from experiments of different RNAi target proteins and cellular fractions were analysed separately. MS/MS data were searched against all entries for *T. brucei* TREU927 listed in the respective fasta file downloaded from the TriTryp database (version 8.1; 11,067), to which the 18 mitochondrially encoded proteins (http://dna.kdna.ucla.edu/trypanosome/seqs/index.html) were added. Database searches were performed with tryptic specificity (with a maximum of two missed cleavages) not allowing cleavage N terminal to proline (Trypsin/P), mass tolerances of 4.5 p.p.m. for precursor and 0.5 Da for fragment ions, carbamidomethylation of cysteine as fixed and N-terminal acetylation and oxidation of methionine as variable modifications. For data from SILAC-labelled samples, Arg10 and Lys8 were set as heavy labels; for dimethyl-labelled samples, dimethylLys0/dimethylNterLys0 and dimethylLys6/dimethylNterLys6 were set as light and heavy labels, respectively. Protein identification by MaxQuant/Andromeda was based on at least one unique peptide with a minimum length of seven amino acids and a false discovery rate of 0.01 applied to both peptide and protein level. For stable isotope-based protein quantification, only unique peptides were considered and the minimum ratio count was set to one. The options 're-quantify' and 'match between runs' were enabled.

In general, only proteins identified with at least two peptides (one of which unique) and quantified in at least two biological replicates per experiment were considered for further bioinformatics and statistical data analyses. For the definition of the ATOM40-dependent mitochondrial importome based on the data set obtained from gradient-purified mitochondria of ATOM40-RNAi cells, we also included proteins identified with only one unique peptide across all replicates if (i) the respective protein was detected and quantified in all four biological replicates and (ii) identification was based on MS/MS spectra (as opposed to identification by 'match between runs') in at least two independent replicates. Annotated MS/MS spectra of single peptide identifications, generated using the Viewer integrated in MaxQuant[48] are shown in Supplementary Data 14. Lists of all proteins identified and quantified in individual data sets are provided in Supplementary Data 1, 3–5, 8 and 10. For all data sets, protein ratios (that is, RNAi $+/-$ Tet) of proteins fulfilling the filter criteria were log$_2$-transformed and mean log$_2$ RNAi $+/-$ Tet ratios determined across all replicates were plotted against the $P$ value calculated for each protein using a two-sided Student's $t$-test.

**Bioinformatics.** *Definition of mitochondrial and glycosomal importomes.* To define mitochondrial and glycosomal importomes, significance thresholds for SILAC ratios demarcating mitochondrial/glycosomal proteins from other proteins were calculated. To this end, we first defined *T. brucei*-specific mitochondrial/glycosomal and non-mitochondrial/non-glycosomal protein reference sets based on literature and entries in the TriTrypDB[49]. The mitochondrial reference proteome consisted of proteins annotated as mitochondrial in the TriTrypDB[49] (version 8.1 as of September 2014) completed by 18 mitochondrially encoded proteins (http://dna.kdna.ucla.edu/trypanosome/seqs/index.html) and proteins reported as mitochondrial in published studies[7,50–61] covering, among others, mitochondrial ribosomes[59], the mitochondrial (outer) membrane proteome[7,50], RNA-editing proteins[55], the mitochondrial RNA-binding complex 1 (MRB1) complex[52,53], mitochondrial carrier proteins[60], the respiratome[51] and the ATPase[61]. This reference proteome was used to compare the data obtained in this study with the set of so far known and published mitochondrial proteins. For the definition of significance thresholds, however, proteins of the OM (except for β-barrel proteins and components of the ATOM complex) were removed from the list since a number of OM proteins do not depend on ATOM40 for membrane insertion and, thus, are not expected to be affected by the ATOM40 knockdown. Details about the mitochondrial reference set comprising 1,238 proteins are provided in Supplementary Data 2a. The non-mitochondrial reference proteome comprising 1,542 proteins included proteins used as such before to define the *T. brucei* MitoCarta[58] as well as proteins of the flagellar proteome[62] and cytosolic ribosomal proteins[63] less proteins with a predicted mitochondrial targeting sequence as determined by TargetP 1.1 (http://www.cbs.dtu.dk/services/TargetP/) (Supplementary Data 2b). The reference proteomes were used to calculate F1 scores integrating the number of true positives (TP), false positives (FP) and

false negatives (FN). The F1 score is defined as harmonic mean of precision [TP/(TP + FP)] and sensitivity [TP/(TP + FN)] and was calculated for given SILAC ratios (mean $\log_2$ values) starting at the lowest ratio at which at least one TP and one FP were present in the data pool and increasing stepwise in 0.01 increments until the maximum ratio was reached. This calculation was performed for both the mitochondrial and the non-mitochondrial reference proteome resulting in distinct F1 scores for each reference proteome. At the ratio giving the maximum F1 score, the number of proteins in the data pool that are present in the reference proteome tested is highest and the number of proteins present in the opposite reference proteome is lowest. The maximum F1 score that imposes the more stringent filter on the data set was set as significance threshold (that is, $t_2 = -0.62$; see Supplementary Fig. 4) to define the mitochondrial importome. Thus, proteins with a SILAC ratio of less than or equal to $-0.62$ and a $P$ value of $<0.05$ were considered significantly decreased in abundance in gradient-purified mitochondria from ATOM40 knockdown compared to control cells and considered as mitochondrial proteins of high confidence.

The reference set for glycosomal proteins was based on a study of the glycosomal proteome for the procyclic form of *T. brucei* and included all glycosomal proteins classified therein as highly confident[38] (Supplementary Data 11a). For the non-glycosomal reference proteome, the set of mitochondrial reference proteins as described above reduced by all entries annotated as glycosomal proteins of high confidence as Güther et al.[38] was used (Supplementary Data 11b). Significance thresholds based on F1 scores were determined as described above for the mitochondrial importome and were as follows: $t_2 = -0.59$ for the glycosomal and $t_1 = -0.33$ for the non-glycosomal reference proteome. Proteins exhibiting a SILAC ratio of less than or equal to $-0.59$ and a $P$ value of $<0.05$ were classified as glycosomal proteins.

*MIA pathway substrate prediction.* Classical MIA pathway substrates present in the genome of *T. brucei* were predicted based on a global Cx3C/Cx9C motif search approach. We searched the genome of *Trypanosoma brucei brucei* TREU927 for proteins ($<50$ kDa) containing two or more Cx3C or Cx9C motifs, which is characteristic for mitochondrial IMS proteins, using the 'Protein Motif Pattern' tool implemented in the TriTrypDB. In addition, a list of 'MIA substrate-like' proteins with sequence or structural similarity to MIA substrates described in other organisms was assembled. This list contains candidates from a domain-based search (similar to Cavallaro[64]), which was performed using the 'InterPro Domain' tool of the TriTrypDB with the following search criteria: organism, *Trypanosoma brucei brucei* TREU927; Pfam domains, PF06747, PF08583, PF05051, PF02297, PF05676 and PF10203; Interpro domain, IPR009069. The list was extended by published tiny TIM-like proteins[31,33] and proteins referred to as homologues of known MIA substrates in the TriTrypDB. For a complete list of the predicted MIA pathway substrates, see Supplementary Data 9.

*Statistical significance test and GO analysis.* For statistical significance testing, we used the two-sided Fisher's exact test to calculate $P$ values. Raw $P$ values were corrected using the Benjamini–Hochberg procedure, which controls the false discovery rate at significance level alpha. In the ERV1 knockdown experiment, proteins with a $P$ value $<0.05$ and a twofold decrease in abundance were considered significantly affected by the ERV1 knockdown. For confirmation, this set of proteins was tested for overrepresentation of predicted MIA substrate-like proteins or Cx3C/Cx9C motif-containing proteins against all proteins quantified in this experiment. Both classes of predicted MIA pathway substrates as well as their combination were highly enriched indicated by $P$ values of $6.05E - 22$ (MIA substrate-like), $6.56E - 18$ (Cx3C/Cx9C motif) and $1.29E - 19$ (combined). GO enrichment analysis of proteins significantly reduced on PEX14 knockdown was based on GO terms provided by the TriTrypDB (version 8.1). For each protein with GO annotation, missing information about ancestor terms were retrieved using the GO.DB R package, lists of GO terms were reduced to GO Slim terms according to the 'GO slim Developed by GO Consortium' database (http://geneontology.org/), and redundant GO terms were removed. The analysis of GO terms that are overrepresented among the glycosomal candidate proteins was performed using the TopGO R package and all proteins quantified in the PEX14-RNAi dataset (Supplementary Data 10) as background. GO terms with a corrected $P$ value of $<0.05$ were considered enriched.

*Blast analysis.* For the identification of yeast (that is, *Saccharomyces cerevisiae*), mouse and human proteins homologous to the new mitochondrial candidate proteins detected in *T. brucei* in this study, a blastp[65] search was performed. The protein sequences of the *T. brucei* candidate proteins were searched against a concatenated database containing the protein sequences present in the Saccharomyces Genome Database (http://www.yeastgenome.org/) and in the UniProt proteome sets for human and mouse (http://www.uniprot.org) as of May 2016 (containing a total of 158,824 entries). The threshold for the expectation value was set to 0.001 and the 10 hits with the highest expectation values were retrieved. In addition, a bitscore threshold of 50 was applied. GO slim terms were assigned to all hits resulting from the blastp search. The results of the Blast analysis are provided in Supplementary Data 7.

*Further bioinformatics tools.* For processing of MaxQuant result files (that is, the 'proteinGroups.txt' file), data visualization, and subsequent statistical and bioinformatics data analyses, an in-house developed software based on R was used including the following R packages: base, data.table, data sets, flux, ggplot2, GO.db, gplots, graphics, grDevices, grid, gtools, methods, pastecs, reshape2, silvermantest, stats, stringr, topGO, utils, VennDiagram and Vennerable. Predictions of N

terminal mitochondrial targeting sequences were performed using TargetP 1.1 (http://www.cbs.dtu.dk/services/TargetP/) and MitoFates (http://mitf.cbrc.jp/MitoFates/cgi-bin/top.cgi). PSORTII (https://psort.hgc.jp/) was used for cellular localization prediction.

**Statistical analysis.** Number of replicates, controls and statistical tests are in accordance with published studies employing comparable techniques and are generally accepted standards in the field. Statistical tests to determine sample size were not employed. To determine proteins significantly affected by RNAi-mediated knockdown of the target protein, a two-sided Student's *t*-test was performed. Protein ratios represent fold change of RNAi-induced samples compared to uninduced control samples. The data obtained were normally distributed, thereby meeting the assumption of the statistical test. Quantitative MS experiments were performed in at least three biological replicates.

**Immunofluorescence microscopy.** Expression of C-terminally triple c-Myc-tagged candidate proteins was induced for 1 day. Cells were fixed with 4% paraformaldehyde in PBS and permeabilized with 0.2% Triton X100 in PBS. Cells were post fixed in cold methanol and slides mounted with VectaShield containing 4′,6-diamidino-2-phenylindole (Vector Laboratories, Product No. H-1200). Images were acquired with a DFC360 FX monochrome camera (Leica Microsystems) mounted on a DMI6000B microscope (Leica Microsystems).

**Northern blotting.** Total RNA was isolated by acid guanidinium thiocyanate-phenol-chloroform extraction[54] from uninduced and induced (3 days) ATOM40-RNAi cells. RNA was separated on a 1% (w/v) agarose gel in 20 mM MOPS/KOH buffer pH 7.0 containing 0.5% (v/v) formaldehyde. Northern probes were generated by PCR from genomic DNA, gel-purified and radioactively labelled using the Prime-a-Gene labelling system (Promega, Product No. U1100). A list of oligonucleotide primers is provided (Supplementary Data 13). Full scans Northern blots are shown in Supplementary Fig. 8.

**Antibodies.** Antibodies used in this study were: mouse anti-c-Myc (Invitrogen, Product No. 132500, dilution 1:2,000 for immunoblotting, 1:50 for immunofluorescence), mouse anti-EF1a (Merck Millipore, Product No. 05-235, 1:10,000), goat anti-mouse Alexa Fluor 633 conjugated (Invitrogen, Product No. A21052, 1:1,000) and goat anti-rabbit FITC conjugated (Sigma, Product No. F0382, 1:1,000); polyclonal rabbit anti-ATOM40 (1:1,000 for both immunoblotting and immunofluorescence), anti-VDAC (1:1,000), anti-COXIV (1:1,000), anti-CYT C1 (1:1,000), anti-TIM9 (1:20) and anti-CYT C (1:100) were previously produced in our laboratory[7,15,44]. Mouse anti-REAP1 (1:1,500)[66], rabbit anti-mtHSP70 (1:1,000)[67], rabbit anti-MCP5 (1:2,500)[68], rabbit anti-LIPDH (1:5,000)[69], rabbit anti-PEX14 (1:1,000)[70], rabbit anti-aldolase (1:50,000)[71], rabbit anti-GAPDH (1:50,000)[72], rabbit anti-G3PDH (1:1,000) and rabbit anti-GIM5 (1:10,000)[37] were kindly provided by L. Krauth-Siegel, S.H. Hajduk, R. Jensen, F. Voncken, P. Michels, A. Jardim and C. Clayton, respectively. Full scans for western blots are shown in Supplementary Fig. 8.

**Miscellaneous.** For the generation of whole-cell extracts, cells were washed with PBS and directly lysed in SDS–PAGE loading buffer. SDS-PAGE and Western blotting of proteins to polyvinylidene difluoride membranes were performed according to standard protocols.

**Data availability.** The MS proteomics data have been deposited to the ProteomeXchange Consortium via the PRIDE[73] partner repository with the data set identifiers PXD004917 (data set: ATOM40 RNAi, gradient-purified mitochondria), PXD005725 (ATOM40 RNAi, whole-cell lysates), PXD004950 (ATOM40 RNAi, organelle-enriched fraction), PXD004979 (SAM50 RNAi), PXD005730 (ERV1 RNAi) and PXD004972 (PEX14 RNAi). The remaining data are available within the article and its Supplementary Information files and from the corresponding authors on reasonable request.

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

## Acknowledgements

We thank the PRIDE team for data deposition to the ProteomeXchange Consortium. Research in the Warscheid group was funded by the Deutsche Forschungsgemeinschaft (grants FOR 1905 and RTG 2202), the Excellence Initiative of the German Federal & State Governments (EXC 294 BIOSS Centre for Biological Signalling Studies) and the European Research Council (ERC, Consolidator Grant Number 648235). Research in the Schneider group was supported by grant 138355 and in part by the NCCR 'RNA & Disease' both funded by the Swiss National Science Foundation.

## Author contributions

J.M. performed molecular genetics, cell biological and biochemical experiments. S.K. performed biochemical fractionations. C.P. performed quantitative MS data analysis, bioinformatics and statistical analysis. A.H. and C.W. generated ERV1-RNAi samples. M.M. performed proteomic experiments of ERV1-RNAi samples and B.K. analysed ATOM40-RNAi whole cell samples. C.P., J.M., A.H. and S.O. prepared figures and tables. All authors designed and analysed the experiments. B.W. and A.S. conceived the project and supervised the study. B.W. and A.S. wrote the manuscript with the input of all other authors. All authors approved the final version of the manuscript.

## Additional information

**Competing interests:** The authors declare no competing financial interests.

