## [Peer Review File · Nature Communications]

PEER REVIEW FILE

Reviewers' Comments:

Reviewer #1 (Remarks to the Author):

This is an interesting paper that, to this reviewer's knowledge, takes a novel approach to determining organelomes, or more specifically organelle importomes.

The principle is simple and elegant: to knockdown a key piece of known organelle protein import machinery (ATOM40 for mitochondrial import and PEX14 for peroxisome/glycosome import) and see which proteins go down in intensity (by SILAC) in preparations enriched for the organelle of interest, compared with non-knockdown controls. An additional component to the analysis (used in the mitochondrial example) is see which proteins go up in the cytoplasmic fraction by virtue of not being imported, though the latter can be confounded by enhanced turnover of the latter.

As with all omic methods, the proof of the pudding is in the eating. Generally, the data (which are well presented) confirm the authors claims as gold standard mitochondrial proteins are highly enriched in their importome, whereas non-mitochondrial proteins are equally excluded and, encouragingly, several mitochondrial protein that should be independent of ATOM40 complex mediated import are unchanged. Based on this fidelity, the authors think they have identified 331 previously unknown mitochondrial proteins (and they further validate several of these by epitope tagging and immunofluorescence). Further, they show that they can determine mitochondrial import using their method even for proteins (like tRNA synthetases) that have dual (mitochondrial and cytoplasmic) localisation, even when the proportion in the mitochondria is low. This is a very nice feature.

Finally, to show that the method can be generalised to other organelles with import machinery, they perform a PEX14 knockdown and look at the proteins that are reduced in intensity in a glycosome-enriched fraction by SILAC (compared with the no-knockdown control) and compare their data with a 'gold standard' high-confidence glycosome proteome determined by a SILAC-based organelle epitope-tagging and pulldown procedure (referred to as reference 36 in supplementary Table 8, but that does not appear in the reference list). In this analysis, they find

61 of the 158 proteins in the high-confidence glycosome. Probably the authors are space-constrained to discuss these data more than they do, but even after removing membrane proteins, they are “missing” many imported glycosome proteins – perhaps indicating that are PEX14-independent import mechanisms (?). Indeed, in the introduction and elsewhere they do not compare their method with the organelle-tag/pulldown quantitative proteomics method that has been used in plants (peroxisomes), toxoplasma (plastid) and *T. brucei* (glycosomes).

In summary, this is a technically well performed study with careful analysis of the data that establishes new methodology to determine organelle “importomes”

Reviewer #2 (Remarks to the Author):

The purpose of this manuscript is essentially to propose a new method to determine the proteome of organelles: to knock down the expression of an essential component of the machinery by which proteins are imported from the cytosol into the organelle and determine the consequent enrichment of proteins in the cytosol. Indeed, this is a novel idea.

The authors apply this approach to the study of two types of organelles in in vitro cultured procyclic (= insect stage) *Trypanosoma brucei*: the mitochondrion (that only occurs as a single large copy in the cell) and glycosomes (present as many small vesicles in a trypanosome). In the mitochondrion, ATOM40 – a translocase in the outer membrane – is depleted by RNAi, in glycosomes PEX14 – a component of the docking complex in the glycosomal membrane participating in the different (PEX5 and PEX7-dependent) matrix protein import pathways. Most of the manuscript (and 4 out of 5 figures) deals with the mitochondrial proteome; the glycosomal study forms only a minor part.

Particularly for the mitochondrial proteome analysis, the authors have described in much detail how they performed all necessary control experiments and made a comparison between the set of proteins predicted as mitochondrial by their accumulation in the cytosol, and a set of well-established mitochondrial reference proteins. In this way, the new method was firmly validated. As the authors demonstrated, this so-called ImportOmics method has several advantages. There is no need for highly purifying organelles and – in contrast to the use of expression of tagged proteins or immunolabelling of protein – is amenable to large-scale proteomics approaches. The method is potent and robust, identifying with high probability a considerable number of new proteins. Moreover, the ImportOmics method will also unambiguously identify proteins as belonging to the organelle under study when such proteins have dual or multiple locations in the cell.

A disadvantage is that proteins will be missed when they are either not synthesized in the nucleus but in the organellar genome (not relevant for organelles without genome such as peroxisomes/glycosomes) and when such proteins are not imported via the depleted transporter (in this work with ATOP40 and PEX4 depletion: e.g. mitochondrial outer membrane and intramembrane space proteins; peroxisomal/glycosomal membrane proteins).

Another disadvantage, is the fact that ImportOmics can only be performed under growth conditions of cells when the level of protein transporter can be considerably decreased, i.e. when the biogenesis of the organelles is not essential. As the authors mentioned, the glycosomes of procyclic trypanosomes could only be studied in the absence of glucose. For the same reason, the method will not be applicable to the human pathogenic stage, the bloodstream-form of trypanosomes. Moreover, various kinds of recent studies with trypanosomes in different labs indicate that the population of glycosomes in a trypanosome is heterogeneous; probably, only part of the population is import competent and the enzyme composition of the organelles may change during growth. Therefore, one has to be careful in referring to THE glycosomal proteome. These latter disadvantages – which will not be restricted to studies with trypanosomes – may require a slightly better mentioning in the paper. Of course, other proteomics procedures, for different organelles and cell types, also measure an average result, but these studies can be done irrespective of the growth conditions.

The work is scientifically sound, the experiments have been skillfully performed and the manuscript has been well written.

Reviewer #3 (Remarks to the Author):

Summary

The manuscript by Peikert et al. reports a new method for organellar proteomics, called 'ImportOmics'. The principle is to knock down a critical component of the organelle's protein import machinery, and to use SILAC based mass spectrometry/bioinformatics to determine which proteins are lost from a subcellular fraction enriched in the organelle. They demonstrate the method with trypanosome mitochondria, for which a good reference proteome and a reliable biochemical enrichment protocol are already available. The method identifies a large proportion of the known mitochondrial trypanosome proteins, as well as many novel candidates, with very low false discovery rate. The authors then provide proof-of-principle data that their method also works with a less well studied organelle, the trypanosome glycosome (a peroxisome related organelle).

Overall assessment

The manuscript is well written, and the method described with good detail. Technically, the study looks very sound; experiments were carefully conducted, with sufficient replicates; the bioinformatics are also solid, and the reasoning for choosing objective significance thresholds is well explained.

I like the method; the approach is straightforward, and seems to work well. I have no substantial criticism of the data. However, I have two principle concerns:

1. The authors claim that this approach will be universally useful for organellar proteomics, but the method is conceptually only suitable for organelles that use a defined import machinery, ie principally mitochondria, peroxisomes, and ER (and so it won't work for endosomes, Golgi, or lysosomes). In the case of ER, it is very questionable if the approach will actually work, as many imported proteins stay only transiently in the ER, and move on to the Golgi, endosomes, lysosomes, and the plasma membrane. I suspect that a block of the ER import machinery would have such drastic consequences for the whole cell that the ImportOmics approach would really struggle to disentangle specific effects. Thus, the method is most suited for organelles that constitute 'end points' of protein trafficking; in combination with the requirement of targetable import machinery, this seems to restrict it to, essentially, mitochondria and peroxisomes. I hence doubt the 'universal' usefulness of the approach (in addition to the technical limitations imposed by having a system amenable to knockdown and SILAC labelling).

2. In the field of organellar proteomics, mitochondria are the easiest target, since they have such distinct fractionation properties. Several recent 'organellar profiling' studies have provided high quality mitochondrial proteomes, without having to resort to knockdown strategies (eg Christoforou et al., Nat Commun, 2016; Itzhak et al., Elife, 2016; Jean Beltran et al., Cell Syst, 2016; even earlier, Forner et al., Mol Cell Proteomics, 2006; Foster et al., Cell, 2006). Thus, the novelty/impact of the approach is somewhat compromised, especially in light of my first criticism above. The glycosome experiment is a proof of principle that another organelle can be targeted, but the data suggest that it worked less well than for mitochondria, and it is not clear what novel insights were gained (other than a non-validated list of candidate glycosome proteins).

Recommendations to enhance the paper's impact

In my opinion, the real strength of the method is that it will allow to dissect the contribution of individual proteins to mitochondrial/peroxisomal import. The authors hint at that possibility with their SAM50 knockdown experiment, but do not fully exploit it. To really make an impact it would be exciting to target a less well studied component of the mitochondrial import machinery, and see what effect that has. The authors have it at their fingertips to use the method to address a novel biological question; I would strongly encourage them to try that.

Furthermore, I have a few more specific comments/suggestions:

L27/28 'An exciting feature of this method is that it overcomes limitations in identifying proteins with dual or multiple locations.'

While this is true, this feature is not entirely novel. For example, the 'protein correlation profiling' approach conceptually copes well with the detection of multiple protein localisations (Foster et al., *Cell*, 2006; although I admit that this method needs to be revisited with up-to-date mass spectrometers). So I would ask the authors to please somewhat tone down claims of novelty for this feature (at several points in the text).

L51 Please also cite the more relevant and recent literature for mitochondrial/organelle proteomics, such as Christoforou et al., *Nat Commun*, 2016; Itzhak et al., *Elife*, 2016; Jean Beltran et al., *Cell Syst*, 2016; Forner et al., *Mol Cell Proteomics*, 2006...

L89/90 The authors show that mitochondrial proteins are slowly depleted, which is later amply corroborated. However, their 'control' for the rest of the proteome is restricted to a Western blot of EF1a. A very important control experiment would be to determine the whole cell proteome from control and KD cells. This will give a global view of which proteins are changing. My worry is that depleting mitochondria may have a more global impact. For example, it is conceivable that a non-mitochondrial protein is depleted indirectly (eg due to altered metabolism, cell death, slowed growth etc). If this protein also partially co-purifies with mitochondria, it will come up as a false positive hit in the Importome.

L119 'pure' mitochondria? Please rephrase, this is misleading.

L167 Although it is useful to show that high-level enrichment is not required, these data show a much reduced sensitivity of the method when partial purification is used (many mitochondrial proteins don't reach significance; Fig S3 vs Fig 2b).

L226 'We retrieved BLAST hits for 91 new mitochondrial candidates and of these, 46 had orthologs in all three species (Fig. 3e), indicating that they are evolutionary conserved.'

Are any of these 46 proteins in the mitochondrial sets of Christoforou et al., *Nat Commun*, 2016; Itzhak et al., *Elife*, 2016; Jean Beltran et al., *Cell Syst*, 2016?

L312 'The majority of the downregulated proteins (35 proteins, 57.4%) are annotated as glycosomal proteins and 12 of these comprise a putative PTS1 according to PSORT II35'

This number is much lower than for the mitochondria. What does that suggest about the FDR of this experiment? Please comment on the quality/FDR of the glycosome experiment.

Reviewer 1

Comment 1:

Finally, to show that the method can be generalised to other organelles with import machinery, they perform a PEX14 knockdown and look at the proteins that are reduced in intensity in a glycosome-enriched fraction by SILAC (compared with the no-knockdown control) and compare their data with a 'gold standard' high-confidence glycosome proteome determined by a SILAC-based organelle epitope-tagging and pulldown procedure (referred to as reference 36 in supplementary Table 8, but that does not appear in the reference list).

Answer: We thank the reviewer for pointing out this mistake. We have corrected it. The paper by Güther et al., 2014 is now included as reference 38 in the reference list.

Comment 2:

In this analysis, they find 61 of the 158 proteins in the high-confidence glycosome. Probably the authors are space-constrained to discuss these data more than they do, but even after removing membrane proteins, they are "missing" many imported glycosome proteins – perhaps indicating that are PEX14-independent import mechanisms. Indeed, in the introduction and elsewhere they do not compare their method with the organelle-tag/pulldown quantitative proteomics method that has been used in plants (peroxisomes), toxoplasma (plastid) and T. brucei (glycosomes).

Answer: In our work, we first and foremost focused on the thorough analysis of the mitochondrial importome and we have even added new data on the identification of novel mitochondrial IMS substrate candidates by depleting the sulfhydryl oxidase ERV1 (see new Fig. 5 and main text, page 13-15).

Our idea of targeting PEX14 was to demonstrate that the method can, in principle, also be applied to glycosomes (see Fig. 6A and 6B) or other organelles with distinct protein import pathways. However, as we pointed out in the text (page 16, second paragraph), our quantitative MS-based analysis of PEX14 knockdown *versus* control cells was only considered as a proof of concept study and represents a minor part of our manuscript.

Unlike in the case of mitochondria, it was not possible to use highly purified organelles for the analysis. The reason is that ablation of PEX14 causes a shift in the density of glycosomes, due to diminished import of proteins. This prevents their isolation by the standard density gradient purification protocol. Instead, we fractionated uninduced and induced PEX14-RNAi cells using digitonin extraction followed by centrifugation. Under these conditions, both "wild-type" and PEX14-depleted glycosome are pelleted. However, we found that glycosomes are only a very minor constituent of the pellets which also contain mitochondria, cytoskeletons, flagella etc. The PEX14 experiment thus illustrates great versatility of our ImportOmics approach and demonstrates that it does not require highly purified organelles. The drawback is that we lose depth of data, which generally increases with the purity of the organelle. We therefore anticipated to cover only a fraction of matrix proteins imported into glycosomes via PEX14. For better clarity, we have added some information and discussion about our glycosomal data in the main text (page 17-18).

Since we consider the PEX14 data as a proof of concept work, we refrained from making comparisons to data obtained by other methods. Moreover, as indicated by the reviewer, it cannot be excluded that alternative PEX14-independent import pathways exist in glycosomes. To address this question in a follow-up ImportOmics study, we envision the generation of highly enriched glycosomal fractions by use of an optimized density gradient purification protocol or by epitope tagging of an abundant glycosomal membrane protein for affinity-purification of glycosomes as reported before (Güther et al., 2014).

To conclude, we are convinced that our new conceptual approach and the data we report here will trigger further studies with the aim to investigate in detail the “importomes” of glycosomes, peroxisomes and plastids as well as the import pathways involved.

Reference

Güther ML, Urbaniak MD, Tavendale A, Prescott A, Ferguson MA. (2014) High-confidence glycosome proteome for procyclic form *Trypanosoma brucei* by epitope-tag organelle enrichment and SILAC proteomics. *J Proteome Res.* 13(6):2796-806.

Reviewer 2

Comment 1:

A disadvantage is that proteins will be missed when they are either not synthesized in the nucleus but in the organellar genome (not relevant for organelles without genome such as peroxisomes/glycosomes) and when such proteins are not imported via the depleted transporter (in this work with ATOP40 and PEX4 depletion: e.g. mitochondrial outer membrane and intramembrane space proteins; peroxisomal/glycosomal membrane proteins).

Answer: The reviewer is correct that mitochondrially encoded proteins do not require import and that these proteins may therefore potentially be missed in our approach. However, we would like to point out here that there are only few of these proteins and that they can be easily predicted from the mitochondrial genome. In *T. brucei*, mitochondrially encoded proteins are notoriously difficult to detect (Škodová-Sveráková et al., 2015). In our importome, we actually detect more mitochondrially encoded proteins (i.e. COII, COIII, MURF2, CYB, A6) than have ever been found before in *T. brucei*. Most mitochondrially encoded proteins are central subunits of larger multiprotein complexes (e.g. respiratory chain complexes I, III, IV and ATP synthase). We found them to be reduced in abundance, most likely as a consequence of a loss of nuclear-encoded partner proteins following ATOM40 knockdown.

Comment 2:

Another disadvantage, is the fact that ImportOmics can only be performed under growth conditions of cells when the level of protein transporter can be considerably decreased, i.e. when the biogenesis of the organelles is not essential. As the authors mentioned, the glycosomes of procyclic trypanosomes could only be studied in the absence of glucose. For the same reason, the method will not be applicable to the human pathogenic stage, the bloodstream-form of trypanosomes.

Answer: We respectfully disagree with the reviewer that ImportOmics can only be performed under growth conditions when the biogenesis of the organelle is not essential. ATOM40, SAM50 and ERV1, which are the targets of the RNAi cell lines used in our mitochondrial ImportOmics experiments, are essential in both the procyclic and the bloodstream form of trypanosomes.

There seems to be a misunderstanding regarding our proof of concept experiment in glycosomes. The PEX14-RNAi cell line employed was in fact grown in the presence of glucose, under conditions when PEX14 is essential for growth and survival of procyclic- and bloodstream-form trypanosomes (see growth curve in Sup. Fig. 6a).

ImportOmics relies on tetracycline-inducible RNAi, a method that is available for many different systems. It allows to choose a time point after RNAi induction when the target protein is efficiently downregulated, but before pleiotropic effects become apparent. For the ATOM40-RNAi cell line, this was shortly after the appearance of the growth phenotype (see growth curve in Supplementary Fig. 1b) and for the newly added ImportOmics experiment using the ERV1-RNAi cell line, this was even before the appearance of the growth phenotype (Supplementary Fig. 6a). Thus, there is no reason why ImportOmics could not be done in the bloodstream form of trypanosomes.

Comment 3:

Moreover, various kinds of recent studies with trypanosomes in different labs indicate that the population of glycosomes in a trypanosome is heterogeneous; probably, only part of the population is import competent and the enzyme composition of the organelles may change during growth. Therefore, one has to be careful in referring to THE glycosomal proteome. These latter disadvantages – which will not be restricted to studies with trypanosomes – may require a slightly better mentioning in the paper. Of course, other proteomics procedures, for different organelles and cell types, also measure an average result, but these studies can be done irrespective of the growth conditions.

Answer: We thank the reviewer for his/her comment. We have included these points in our revised version of the manuscript (see new text on page 17 and 18). We would to add that our method relying on RNAi-mediated knockdown of a central import component induced by tetracycline can be performed at different growth conditions.

Reference

Škodová-Sveráková I, Horváth A, Maslov DA. (2015) Identification of the mitochondrially encoded subunit 6 of F1FO ATPase in *Trypanosoma brucei*. *Mol Biochem Parasitol.* 201:135-8.

Reviewer 3**Comment 1:**

The authors claim that this approach will be universally useful for organellar proteomics, but the method is conceptually only suitable for organelles that use a defined import machinery, ie principally mitochondria, peroxisomes, and ER (and so it won't work for endosomes, Golgi, or lysosomes). In the case of ER, it is very questionable if the approach will actually work, as many imported proteins stay only transiently in the ER, and move on to the Golgi, endosomes, lysosomes, and the plasma membrane. I suspect that a block of the ER import machinery would have such drastic consequences for the whole cell that the ImportOmics approach would really struggle to disentangle specific effects. Thus, the method is most suited for organelles that constitute 'end points' of protein trafficking; in combination with the requirement of targetable import machinery, this seems to restrict it to, essentially, mitochondria and peroxisomes. I hence doubt the 'universal' usefulness of the approach (in addition to the technical limitations imposed by having a system amenable to knockdown and SILAC labelling).

Answer: We agree with the reviewer that our method is best suited for the study of organelles that constitute "end points" of protein trafficking. We also agree that the ER might not be the most suitable organelle to be studied by ImportOmics. Nonetheless, our ImportOmics approach should be, in addition to mitochondria, well applicable to the study of mitochondria-related organelles such as hydrogenosomes and mitosomes (van der Giezen et al., 2005). Many of these organelles are difficult to purify and may therefore be especially suited for ImportOmics which does not require highly purified organelles. Most importantly, ImportOmics should be very useful to study chloroplasts and other plastids. In the case of plastids, it might even be possible to assign substrates to the different thylakoid protein import systems. In plant and algae systems, comparative ImportOmics experiments targeting mitochondria and plastids can be envisioned. Finally, it should in principle be possible to study nuclear protein import by ablating the Ran-GTPase and/or nuclear transport receptors. We have added this information to the Discussion section on page 18/19 in the revised version of the manuscript.

The reviewer is correct that ImportOmics (at least in the case where essential proteins are targeted) is only possible in systems in which inducible knockdowns are possible. However, considering the technological advancements made over recent years concerning the application of RNAi and

CRISPR/Cas9 technology, this is not of such concern anymore as more and more experimental systems will be amenable to conditional knockdowns.

Finally, the reviewer mentions the important point that ImportOmics as presented in the original manuscript was used in combination with the SILAC methodology. However, as shown in the revised manuscript our method is not restricted to SILAC and, thus, organisms to which SILAC can be applied. ImportOmics can also be readily combined with chemical stable isotope labeling (e.g. dimethyl labeling, TMT labeling) or even label-free MS approaches. Thus, it is directly applicable to organisms that cannot be grown in defined media and, therefore, are refractory to SILAC or cells in which complete SILAC labeling is difficult to achieve.

In the ImportOmics experiment that targets ERV1, which is presented in the revised manuscript (see Fig. 5, Supplementary Fig. 6, and Supplementary Table 8), SILAC was successfully replaced by peptide stable isotope dimethyl labeling (Boersema et al., 2009) (see also Methods “Stable isotope dimethyl labeling”). This method does not rely on metabolic incorporation of stable isotopes; instead, proteolytic peptides are chemically labeled by stable isotopes for relative quantitative MS analysis. The exciting results obtained by the ERV1 experiment illustrate that ImportOmics does not obligatorily rely on SILAC but that it can straightforwardly be performed using postlabeling methods, further adding to its versatility.

In summary, while we agree with the reviewer that ImportOmics is not a "universal" method (which I don't think is claimed in the manuscript), we hope that the arguments laid out above convincingly show that ImportOmics is in principle widely applicable to various organelles in different organisms.

Comment 2:

In the field of organellar proteomics, mitochondria are the easiest target, since they have such distinct fractionation properties. Several recent ‘organellar profiling’ studies have provided high quality mitochondrial proteomes, without having to resort to knockdown strategies (eg Christoforou et al., Nat Commun, 2016; Itzhak et al., Elife, 2016; Jean Beltran et al., Cell Syst, 2016; even earlier, Forner et al., Mol Cell Proteomics, 2006; Foster et al., Cell, 2006). Thus, the novelty/impact of the approach is somewhat compromised, especially in light of my first criticism above.

Answer: We are aware of these studies. First, we would like to point out that in none of the studies mentioned, trypanosomes have been studied. In previous work, we already successfully employed protein profiling approaches to characterize the proteome of human liver and mouse kidney peroxisomes (Gronemeyer et al. 2013, Wiese et al., 2007) as well as to delineate the mitochondrial outer membrane proteome in *T. brucei* (Niemann et al. 2013). The studies by Christoforou et al. (2016), Itzhak et al. (2016), Jean Beltran et al. (2016) and Foster et al (2006) impressively show that protein profiling approaches allow for the establishment of organellar maps on a global scale. However, these global approaches are still limited in depth when looking at the number of proteins localized to a distinct organelle. This is even the case for mitochondria which arguably are the “easiest target”. Foster et al. (2006) reports localization of 297 proteins to mouse live mitochondria. Christoforou et al. (2016) localizes 585 protein to mitochondria in mouse ES cell and Itzhak et al. (2016) 658 proteins to mitochondria in HeLa cells. Since mammalian mitochondria comprise more than 1,000 (estimated size is up to 1,500 mitochondrial proteins), it is apparent that these studies cover only a fraction of the mitochondrial proteome. In our study, we report a most comprehensive importome of mitochondria in the unicellular *T. brucei* by successfully localizing 1,120 proteins including 331 new proteins to this important organelle.

The reviewer is correct with the point that organellar profiling approaches do not require knockdown strategies. However, we see advantages in our ImportOmics method as data acquisition and analysis is fast and much easier in comparison to global protein profiling approaches which are based on the analysis of many density gradient fractions (leading to the generation of very large MS data sets which require thorough quantitative and statistical analysis by advanced computational approaches). Moreover, ImportOmics in contrast to most other methods that are used to determine organellar

proteomes does not require gradient-purified organellar fractions. The SAM50 and ERV1 ImportOmics experiments, as well as the PEX14 ImportOmics experiment for glycosomes, illustrate that very good results are obtained using extremely crude organellar fractions only.

Furthermore, the ImportOmics method allows for both the study of protein inventories of organelles (importomes) as well as to define substrates of distinct protein import pathways. To better demonstrate this unique feature and the high potential of our method, we have now included in the revised version of the manuscript novel and very interesting data on the mitochondrial intermembrane space import and assembly (MIA) pathway (for details, please refer to our answer to comment 4).

In conclusion, we respectfully disagree with the reviewer that the novelty or impact of our method is compromised in light of previous studies including global organellar profiling approaches. We rather think that our approach is distinct from organellar profiling methods and that it opens up new avenues for the study of organellar proteomes and the protein import pathways involved. Since the method is fast, easy, and precise, we strongly believe that cell biology labs can readily apply our ImportOmics method and adapt it to their specific question.

Comment 3:

The glycosome experiment is a proof of principle that another organelle can be targeted, but the data suggest that it worked less well than for mitochondria, and it is not clear what novel insights were gained (other than a non-validated list of candidate glycosome proteins).

Answer: The reviewer is correct with the statement that the glycosome experiment is a proof of concept to show that our method is not only applicable to mitochondria but also to other organelles with a defined import machinery such as peroxisomes, chloroplasts or other plastids. This part of the work was therefore by no means intended to present an in-depth characterization of glycosomes or to provide novel insight with regard to the identification of new glycosomal matrix proteins and, thus, represents only a minor part of our manuscript. In this proof of concept experiment, we only applied a fast and very simple cell extraction method using digitonin, which yields a crude organelle-enriched fraction comprising glycosomes in addition to all other organelles (based on total MS intensities, we estimate an abundance of less than 10% for glycosomes in this fraction). It is also important to note here that the depth of data increases with the purity of the organelle fraction analyzed (see Fig. 2b and Supplementary Fig. 3a and 3b), a limitation that commonly applies to organellar proteomics approaches. Thus, based on the data we could obtain from this very crude organellar fraction, we do not think that it can be concluded that the method works less well for glycosomes. In fact, we see a clear effect on glycosomal enzymes involved in glycolysis, while PEX proteins are not affected by PEX14 knockdown as highlighted in Fig. 6c. However, there may be different reason why only a fraction of known glycosomal enzymes was covered and found to be affected: First, as discussed above, the depth and unambiguousness of data increases with the purity of the organelle fraction used for the analysis. In case of glycosomes, non-imported glycosomal proteins are not degraded in the cytosol as it is the case for mitochondrial proteins (see Fig. 1e and Fig. 6b). Thus, a rather minor cytosolic contamination of the organelle-fraction (see Fig. 6b, EF1a) may mask the down-regulation of some glycosomal enzymes inside the glycosomal matrix compartment in our SILAC-MS analysis. Second, glycosomes are highly dynamic and heterogeneous organelles and most likely import different sets of enzymes depending on their metabolic state and the environmental condition. Third, so far it is not known whether all glycosomal matrix proteins require PEX14 for import or whether some glycosomal enzymes may even be imported independently of PEX14. To better address these points, we have added some information and discussion about our glycosomal data in the main text (page 17 and 18).

Comment 4:

In my opinion, the real strength of the method is that it will allow to dissect the contribution of individual proteins to mitochondrial/peroxisomal import. The authors hint at that possibility with their SAM50 knockdown experiment, but do not fully exploit it. To really make an impact it would be

exciting to target a less well studied component of the mitochondrial import machinery, and see what effect that has. The authors have it at their fingertips to use the method to address a novel biological question; I would strongly encourage them to try that.

Answer: We thank the reviewer for his/her recommendation to enhance the paper's impact. We have followed the suggestion and have targeted the mitochondrial disulfide relay system involved in the import of the majority of soluble intermembrane space (IMS) proteins. In yeast and human, the mitochondrial intermembrane space assembly (MIA) pathway requires the oxidoreductase MIA40 and the sulfhydryl oxidase ERV1/ALR. Interestingly, *T. brucei* expresses an orthologue of ERV1 but lacks a MIA40, which also raises the question how ERV1 functions and whether it is sufficient to mediate import of IMS proteins. Furthermore, experimental evidence for trypanosomal MIA pathway substrates is lacking. To globally identify substrates of the import factor ERV1, we performed ImportOmics experiments using ERV1-RNAi cells as described on pages 13-15 (see also Fig. 5, Supplementary Fig. 6, and Supplementary Table 8). As a result, we report for the first time a valid set of 25 putative MIA pathway substrates that require ERV1 function for import into mitochondria. With this approach, we are also able to identify non-classical MIA pathway substrates that do not comprise twin-Cx3C or Cx9C motives (with x being any amino acid except cysteine) and/or typical MIA pathway substrate domains. Our data therefore demonstrate the high potential and effectiveness of ImportOmics for the identification of *in vivo* substrates of distinct protein import factors of organelles.

Comment 5:

L27/28 'An exciting feature of this method is that it overcomes limitations in identifying proteins with dual or multiple locations.'

While this is true, this feature is not entirely novel. For example, the 'protein correlation profiling' approach conceptually copes well with the detection of multiple protein localisations (Foster et al., Cell, 2006; although I admit that this method needs to be revisited with up-to-date mass spectrometers). So I would ask the authors to please somewhat tone down claims of novelty for this feature (at several points in the text).

Answer: We did not intend to claim that this feature is entirely novel and we have referenced powerful protein profiling approaches that provide the potential to identify proteins with dual/multiple localizations (page 3). However, we still see challenges in accurately localizing proteins with multiple localizations especially when only a very small fraction of such a protein is localized in the organelle as exemplified by aminoacyl-tRNA synthetases in the manuscript. For better clarity, we have toned down the respective text (see abstract, for example) in the revised manuscript as suggested by the reviewer.

Comment 6:

L51 Please also cite the more relevant and recent literature for mitochondrial/organelle proteomics, such as Christoforou et al., Nat Commun, 2016; Itzhak et al., Elife, 2016; Jean Beltran et al., Cell Syst, 2016; Forner et al., Mol Cell Proteomics, 2006...

Answer: This has been done (see page 3).

Comment 7:

L89/90 The authors show that mitochondrial proteins are slowly depleted, which is later amply corroborated. However, their 'control' for the rest of the proteome is restricted to a Western blot of EF1a. A very important control experiment would be to determine the whole cell proteome from control and KD cells. This will give a global view of which proteins are changing. My worry is that depleting mitochondria may have a more global impact. For example, it is conceivable that a non-mitochondrial protein is depleted indirectly (eg due to altered metabolism, cell death, slowed growth etc). If this protein also partially co-purifies with mitochondria, it will come up as a false positive hit in the Importome.

Answer: The reviewer is correct that we have not included data showing changes in the whole cell proteome of induced (+Tet) in comparison to uninduced (-Tet) ATOM40-RNAi cells. We have now added these data which are presented in the new Supplementary Fig. 3a and Supplementary Table 3. The new dataset is described in the main text on page 8 and 9 and the experimental details are included in the Methods section. Please note that the ATOM40-RNAi cells used for this experiment were taken from the same cultures that were used to isolate mitochondria for defining the mitochondrial importome. In our SILAC-MS analysis of whole cell extracts of tetracycline-induced *versus* uninduced ATOM40-RNAi cells, we have relatively quantified 5,144 proteins. We considered proteins with a fold-change of 1.5 and a p-value < 0.05 (n=3) to be significantly regulated. Based on the analysis of mitochondrial and non-mitochondrial reference proteomes (Supplementary Fig. 3a), we conclude that loss of ATOM40 specifically diminished mitochondrial proteins, while non-mitochondrial proteins remained virtually unaffected. Thus, we have no indication that non-mitochondrial proteins, if partially co-purified with mitochondria, represent a source of false positive hits in our ImportOmics study. In fact, we think that the new data underscore the specificity of our approach for determining the mitochondrial importome.

Comment 8:

L119 *'pure' mitochondria? Please rephrase, this is misleading.*

Answer: For better clarity, we have removed 'pure' or replaced it by 'gradient-purified' throughout the manuscript.

Comment 9:

L167 *Although it is useful to show that high-level enrichment is not required, these data show a much reduced sensitivity of the method when partial purification is used (many mitochondrial proteins don't reach significance; Fig S3 vs Fig 2b).*

Answer: We agree with the reviewer that the sensitivity of the approach is certainly enhanced when mitochondrial fractions of higher or high purity are analyzed as stated in the manuscript on page 18. However, we would like to point out that the 'crude' mitochondrial fractions analyzed in our study (Supplementary Fig. 3b) were obtained by a single step cell fractionation method using digitonin, by which an organelle-enriched pellet fraction is obtained. Thus, mitochondria are neither specifically enriched in this sample nor separated from most other organelles. With this information, we think that the data nicely show the potential of our ImportOmics approach for the study of organelles which have not been purified by density gradient centrifugations.

Comment 10:

L226 *'We retrieved BLAST hits for 91 new mitochondrial candidates and of these, 46 had orthologs in all three species (Fig. 3e), indicating that they are evolutionary conserved.'*

Are any of these 46 proteins in the mitochondrial sets of Christoforou et al., Nat Commun, 2016; Itzhak et al., Elife, 2016; Jean Beltran et al., Cell Syst, 2016?

Answer: We have checked the 46 proteins for their identification and mitochondrial localization in recent studies mentioned by the reviewer (see also new sentence on page 11/12 in the revised manuscript). Of the 46 proteins, 42 (Christoforou et al., 2016), 46 (Itzhak et al., 2016), and 29 (Jean Beltran et al. 2016) were reported to have a mitochondrial localization.

Comment 11:

L312 *'The majority of the downregulated proteins (35 proteins, 57.4%) are annotated as glycosomal proteins and 12 of these comprise a putative PTS1 according to PSORT II35'*

This number is much lower than for the mitochondria. What does that suggest about the FDR of this experiment? Please comment on the quality/FDR of the glycosome experiment.

Answer: We would like to point out that the glycosomal proteome is also much smaller and estimated to be about one fifth of the mitochondrial proteome with regard to the number of individual components. Furthermore, many glycosomal proteins are of low abundance. In fact, glycosomal proteins (i.e. proteins of the glycosomal reference proteome) only account for less than 10% of the overall MS intensity measured in the organelle-enriched fractions used for this experiment. Thus, 90% of the MS intensity originates from "contaminants". In addition, the glycosomal reference proteome is certainly less robust than the mitochondrial reference set, because glycosomes have been less well studied. We therefore refrain from calculating an FDR for this experiment. However, if calculated based on our glycosomal and non-glycosomal reference sets, the FDR for the set of significantly downregulated proteins is 0.125. Nevertheless, as pointed out above (see reply to comment 3), this experiment was meant as proof of concept to demonstrate that our method is also applicable to other organelles. Considering the very low abundance of glycosomal proteins in the samples analyzed, we believe that the overall quality of the glycosome experiment is fairly good.

References

- Boersema, P.J., Raijmakers, R., Lemeer, S., Mohammed, S., Heck, A.J. (2009) Multiplex peptide stable isotope dimethyl labeling for quantitative proteomics. *Nat Protoc* 4(4):484-94.
- Christoforou, A., Mulvey, C.M., Breckels, L.M., Geladaki, A., Hurrell, T., Hayward, P.C., Naake, T., Gatto, L., Viner, R., Martinez Arias, A., Lilley, K.S. (2016) A draft map of the mouse pluripotent stem cell spatial proteome. *Nat Commun* 7, 8992.
- Gronemeyer, T., Wiese, S., Ofman, R., Bunse, C., Pawlas, M., Hayen, H., Eisenacher, M., Stephan, C., Meyer, H.E., Waterham, H.R., Erdmann, R., Wanders, R.J.A., Warscheid, B. (2013) The proteome of human liver peroxisomes: Identification of five new peroxisomal constituents by a label-free quantitative proteomics survey. *PLoS One* 8(2):e57395.
- Itzhak, D.N., Tyanova, S., Cox, J. & Borner, G.H. (2016) Global, quantitative and dynamic mapping of protein subcellular localization. *Elife* 5.
- Jean Beltran, P.M., Mathias, R.A. & Cristea, I.M. (2016) A Portrait of the Human Organelle Proteome In Space and Time during Cytomegalovirus Infection. *Cell Syst* 3, 361-373 e366.
- Niemann, M., Wiese, S., Mani, J., Chanfon, A., Jackson, C., Meisinger, C., Warscheid, B.*, Schneider, A.* (2013) Mitochondrial outer membrane proteome of *Trypanosoma brucei* reveals novel factors required to maintain mitochondrial morphology. *Mol Cell Proteomics* 12(2):515-528.
- van der Giezen, M., Tovar, J., Clark, C.G. (2005) Mitochondrion-derived organelles in protists and fungi. *Int Rev Cytol* 244:175-225.
- Wiese, S., Gronemeyer, T., Ofman, R., Kunze, M., Grou, C.P., Almeida, J.A., Eisenacher, M., Stephan, C., Hayen, H., Schollenberger, L., Korosec, T., Waterham, H.R., Schliebs, W., Erdmann, R., Berger, J., Meyer, H.E., Just, W., Azevedo, J.E., Wanders, R.J.A., Warscheid, B. (2007) Characterization of mouse kidney peroxisomes by tandem mass spectrometry and protein correlation profiling. *Mol Cell Proteomics* 6(12):2045-2057.

Reviewers' Comments:

Reviewer #2 (Remarks to the Author):

The authors have appropriately dealt with the comments on the initial version.

I should mention that there has been a misunderstanding of my (reviewer 2) comment 2, possibly because the comment should have been more accurately formulated (for which my apology). The point I wished to make is that it has been shown by studies in different laboratories that mislocalization of even a minor amount of glycosomal matrix proteins in bloodstream-form *T. brucei* results almost immediately in growth arrest followed rapidly by death. It has not been possible yet to find growth conditions for these bloodstream-forms which allow longer term survival with mislocalized glycosomal proteins. Therefore, I remain in doubt whether ImportOmics by RNAi of peroxins can be performed with these cells. This problem does not apply to procyclic-form *T. brucei*, where the onset of growth arrest upon RNAi of peroxins occurs much later (and indeed, the authors did perform such experiments). Obviously, I don't (and didn't) question the possibility to apply ImportOmics for studying mitochondrial proteome of these bloodstream cells. There is no need to discuss this further or address in in the manuscript; in my opinion, the manuscript is fine to be published as it stands.

Reviewer #3 (Remarks to the Author):

The revised version of the manuscript by Peikert et al. is much improved over the first submission. Following my recommendations, the authors have performed an important additional control experiment (quantitative proteomics of cell lysates in ATOM40 knockdown vs. control cells), and have also added new biological insights by applying their method to the MIA pathway. All minor comments have also been addressed. Hence, in my opinion, the manuscript is now ready for publication.

I have one more question though. The new Supplementary Figure 6c shows reasonably high correlation of SILAC ratios between the replicates (around 0.6). The MIA pathway inactivation only affects a handful of proteins (25), as the plot clearly shows. I would therefore expect to see almost no correlation between repeats, since most SILAC ratios should be unaffected (and scatter around 1). Surprisingly, many proteins appear to have reproducible large positive ratios (>4 in some cases; see also Supplementary Table 8). This contrasts with Supplementary Figure 2, where large positive ratios are scarce, and the correlation is mostly driven by genuine depletion (large negative ratios) of mitochondrial proteins. In the new Figure 5c, only the negative ratios

are shown; including the right part of the plot would show an unexpected large cloud of proteins with positive ratios.

While addressing this question should not delay publication of the manuscript, I would be grateful if the authors could share their thoughts on this observation with me.

Dear Dr. Larochelle,

Please find below our answers to the latest comments of Reviewer 2 and Reviewer 3:

Reviewer 2

Comment:

The authors have appropriately dealt with the comments on the initial version.

*I should mention that there has been a misunderstanding of my (reviewer 2) comment 2, possibly because the comment should have been more accurately formulated (for which my apology). The point I wished to make is that it has been shown by studies in different laboratories that mislocalization of even a minor amount of glycosomal matrix proteins in bloodstream-form *T. brucei* results almost immediately in growth arrest followed rapidly by death. It has not been possible yet to find growth conditions for these bloodstream-forms which allow longer term survival with mislocalized glycosomal proteins. Therefore, I remain in doubt whether ImportOmics by RNAi of peroxins can be performed with these cells. This problem does not apply to procyclic-form *T. brucei*, where the onset of growth arrest upon RNAi of peroxins occurs much later (and indeed, the authors did perform such experiments). Obviously, I don't (and didn't) question the possibility to apply ImportOmics for studying mitochondrial proteome of these bloodstream cells. There is no need to discuss this further or address it in the manuscript; in my opinion, the manuscript is fine to be published as it stands.*

Answer: We would like to thank the reviewer for this clarification and providing further information on the growth behavior of bloodstream-form *T. brucei* when import of glycosomal matrix proteins is impaired. This is indeed an important observation and it will be of great interest to test whether ImportOmics can be applied in this specific case. However, in all our experiments we carefully made sure that cells were harvested at or even before the onset of a growth arrest. Therefore, we still see the possibility of applying the ImportOmics approach for the study of glycosomal import processes in bloodstream-form *T. brucei*.

Reviewer 3

Comment:

The revised version of the manuscript by Peikert et al. is much improved over the first submission. Following my recommendations, the authors have performed an important additional control experiment (quantitative proteomics of cell lysates in ATOM40 knockdown vs. control cells), and have also added new biological insights by applying their method to the MIA pathway. All minor comments have also been addressed. Hence, in my opinion, the manuscript is now ready for publication.

I have one more question though. The new Supplementary Figure 6c shows reasonably high correlation of SILAC ratios between the replicates (around 0.6). The MIA pathway inactivation only affects a handful of proteins (25), as the plot clearly shows. I would therefore expect to see almost no correlation between repeats, since most SILAC ratios should be unaffected (and scatter around 1). Surprisingly, many proteins appear to have reproducible large positive ratios (>4 in some cases; see also Supplementary Table 8). This contrasts with Supplementary Figure 2, where large positive ratios are scarce, and the correlation is mostly driven by genuine depletion (large negative ratios) of mitochondrial proteins. In the new Figure 5c, only the negative ratios are shown; including the right part of the plot would show an unexpected large cloud of proteins with positive ratios.

While addressing this question should not delay publication of the manuscript, I would be grateful if the authors could share their thoughts on this observation with me.

Answer: We would like to thank the reviewer for her/his positive response. We are also happy to further comment on the quantitative proteomics data of ERV1 knockdown. The reviewer is absolutely correct that only a distinct set of proteins was significantly reduced in abundance, while at the same time a number of proteins exhibited increased ratios. We did not observe such an increase in protein abundance following ATOM40 knockdown, which may indicate that this effect is specific for ERV1 knockdown leading to the accumulation of MIA substrates in the cytosol. At this point it is important to note that in the experiment shown in Supplementary Figure 6c, we analyzed organelle-enriched fractions of induced *versus* uninduced ERV1-RNAi cells.